



# A novel method of identifying and analysing oil smoke plumes based on synergic satellite data

Alexandru Mereuţă[1], Nicolae Ajtai[1*], Andrei T. Radovici[1], Nikolaos Papagiannopoulos[2], Lucia T. Deaconu[1], Camelia S. Botezan[1], Horaţiu I. Ştefănie[1,3], Doina Nicolae[4], Alexandru Ozunu[1,5]

[1]Faculty of Environmental Science and Engineering, Babeş-Bolyai University, Cluj-Napoca, 400294, Romania
[2]Consiglio Nazionale delle Ricerche, Istituto di Metodologie per l'Analisi Ambientale (CNR-IMAA), C.da S. Loja, Tito Scalo (PZ), 85050, Italy
[3]Institute of Geophysics, Faculty of Physics, University of Warsaw, Warsaw, Poland
[4]National Institute of R&D for Optoelectronics (INOE), Magurele, Romania
[5]Faculty of Natural and Agricultural Sciences, Disaster Management Training and Education Centre (DIMTEC), University of the Free State, Bloemfontein 9300, South Africa

*Correspondence to:* Nicolae Ajtai (nicolae.ajtai@ubbcluj.ro)

**Abstract.** Black carbon aerosols are the second largest contributor to global warming while also being linked to respiratory and cardiovascular disease. These particles are generally found in smoke plumes originating from biomass burning and fossil fuel combustion. They are also heavily concentrated in smoke plumes originating from oil fires exhibiting the largest ratio of black carbon to organic carbon. In this study, we identified and analyzed oil smoke plumes derived from 30 major industrial events within a 12-year timeframe. To our knowledge, this is the first study of its kind that utilized a synergetic approach based on satellite remote sensing techniques. One objective of this study is to highlight the importance of satellite remote sensing techniques in identifying these types of events. As opposed to ground stations, satellite data offers access to remote areas all over the globe which would otherwise be very difficult to reach. Satellite data offers access to these events which, as seen in this study, are mainly located in war prone or hazardous areas. This study focuses on the use of MODIS (Moderate Resolution Imaging Spectroradiometer) and CALIPSO (Cloud-Aerosol Lidar and Infrared Pathfinder Satellite Observations) products regarding these types of aerosol while also highlighting their intrinsic limitations. By using data from both MODIS instruments onboard Terra and Aqua satellites we addressed the temporal evolution of the smoke plume while assessing Lidar specific properties and plume elevation using CALIPSO data. We present several aerosol properties in the form of plume specific averaged values. The MODIS ocean algorithms were successful in retrieving aerosol properties which, on average, ranged from -0.06 to 0.16 for plume specific AOD, - 0.18 to 1.25 for Ångström exponent and 0.29 to 1.73 μm for the effective radius. CALIPSO measurements showed values of plume AOD ranging from 0 to 0.14 (532 nm) and 0 to 0.13 (1064 nm) except for one event where AOD values showed 1.52 (532 nm) and 1.42 (1064 nm). AE values ranged from 0.11 to 0.33 which were in agreeance with MODIS values. A large discrepancy can be found in one event where CALIPSO measured AOD values 5 times higher than MODIS. This event also produced the largest lidar ratio at 109 sr (532 nm) and 86 (1064 nm). Other lidar ratio values ranged from 37 to 55 sr however these unconstrained solutions were obtained for the entire layer of which the plumes were a part of and thus did not reflect specific plume conditions. Particulate backscatter



values ranged from 0.002 to 0.0017km$^{-1}$ sr$^{-1}$ while extinction coefficient values ranged from 0.10 to 1.65 km$^{-1}$. On average

backscatter and extinction coefficient values were 2 to 9 times higher than local background values. Particulate depolarization ratios ranged from 0.11 to 0.15 in 4 out of 6 cases while the remaining two ranged from 0.27 to 0.32 where dust was highly dominant. The values represented in this study are in good agreement with similar studies that used ground based and flight measurements. We believe that MODIS values are a conservative estimation of plume AOD since MODIS algorithms rely on general aerosol models and various atmospheric conditions within the look-up tables which do not reflect

the highly absorbing nature of these smoke plumes. CALIPSO measurements are heavily dependent on lidar ratios which are not directly measured if plumes within the planetary boundary layer. We also believe that AOD values based on CALIPSO measurements are conservative in nature since heavy absorbing smoke would yield larger lidar ratios and AOD values. Based on this study we conclude that the MODIS land algorithms are not yet suited for retrieving aerosol properties for these types of smoke plumes due to the strong absorbing properties of these aerosols. We believe that these types of studies are a

strong indicator for the need of improved aerosol models and retrieval algorithms.

## 1 Introduction

Atmospheric aerosols are chemically complex mixtures of solid and liquid particles dynamically suspended in air. They originate from both natural and anthropogenic emissions. More common naturally occurring aerosols can be observed in the form of fog, dust, sea salt spray, biological exudates and grey smoke (biomass burning). Haze, smog and black smoke

are typically a result of industrial and transportation activities (Stocker et al., 2014; Wei et al., 2020). Distinct species such as black carbon (BC), organic carbon (OC), sulphates, nitrates, trace elements, sea salt, mineral dust and biological matter suffer atmospheric alteration resulting in different combinations of compounds. Defining aerosol types is a difficult task as they possess a large degree of variance in composition and concentration due to different atmospheric residence times, dry deposition and wet scavenging, emission rates and sources, transport trajectories and seasonal variability (Dutkiewicz et al.,

2009; Li et al., 2015; Samset et al., 2018).

Health effects associated with both short and long-term exposure to aerosol have been widely documented in scientific literature (Brauer et al., 2015; Laumbach and Kipen, 2012; Zhang and Batterman, 2013; Pascal et al., 2013; Lee et al., 2014; Guarnieri and Balmes, 2014; Zhang et al., 2017). Aerosols have been linked to respiratory and cardiovascular diseases due to fine particulate matter (PM2.5; <2.5 μm in diameter) that can penetrate the lungs resulting in increased rates

of morbidity and mortality (Brunekreef and Holgate, 2002; Pope III et al., 2002; Lim et al., 2012; Beelen et al., 2014; Hoek et al., 2013).

Global circulation of aerosols is a known transport vector of minerals and nutrients to the biosphere (McTainsh and Strong, 2007; Maher et al., 2010; Lequy et al., 2012). Aerosols have a direct effect on radiation distribution by scattering, absorbing and emitting light thorough the atmosphere. In addition they can affect the climate system through indirect effects

acting as cloud condensation nuclei, impacting cloud lifetime and properties, atmospheric stability and precipitation factors



(Popp et al., 2016; Samset et al., 2018; Stocker et al., 2014). They can disrupt circulation patterns, impact air temperatures and severe weather systems (Fan et al., 2016), reduce visibility (Deng et al., 2012; Wang et al., 2015) and lead to ozone depletion (Popp et al., 2016).

Because of their complex influence on the environment and climate system assessing aerosol key parameters is essential for any atmospheric study. Aerosol optical depth (AOD), the extinction vertically integrated throughout the atmospheric column, is strongly correlated to PM concentrations. Together with other properties such as Ångström exponent (AE), single scattering albedo (SSA), size distribution and vertical distribution we can better describe their atmospheric impacts. Between ground stations and spaceborne observations, satellite remote sensing offers a more comprehensive global view of aerosols. Since the 1970s onwards, there has been a significant number of satellite sensors used successfully for

retrieving AOD and other aerosol properties (Li et al., 2015, 2016; Dubovik et al., 2019; Schutgens et al., 2020; Wei et al., 2020; Sayer et al., 2020). When choosing between different aerosol products one must take into account the wide variety of sensors and their characteristics such as spatial, temporal and spectral resolutions, single or multi-view retrieval methods, intensity or polarimetric design and different retrieval algorithms (Wei et al., 2020; Fan and Qu, 2019; Sogacheva et al., 2020; Li et al., 2020). In addition to sensor characteristics, other factor such as cloud coverage, surface type, aerosol models

and retrieval algorithms contribute to overall retrieval uncertainties (Wei et al., 2020; Li et al., 2015; Virtanen et al., 2018). Most sensors cannot retrieve a wide variety of aerosol properties thus relying on inversions techniques and complex radiative transfer computations (Schutgens et al., 2020).

Smoke aerosols are primarily composed of two distinct carbonaceous species: BC, highly absorbent in all visible wavelengths and OC highly scattering  of solar radiation (Ramanathan and Carmichael, 2008; Dutkiewicz et al., 2009).

Recent studies suggest black carbon (BC) is the second largest contributor to global warming after $CO_2$, however their impact on global radiative budget is still a subject of debate posing significant uncertainties (Bond et al., 2013; Andreae and Ramanathan, 2013; Stocker et al., 2014; Boucher et al., 2016). BC (or soot) is generated from incomplete combustion processes with 59% accounting from biomass burning while the rest is attributed to fossil fuel combustion (Bond et al., 2013). Particle morphology of BC is significantly affected by multi-phase processes shortly after being emitted

(Kokhanovsky, 2019; Noyes et al., 2020). Aging/coating processes further change optical and microphysical properties of BC with direct impacts on their radiative effects (Riemer et al., 2010; He et al., 2015; Fierce et al., 2015; Peng et al., 2016). In practice, many remote sensing studies use rough estimations of particle shapes for fresh and aged BC further inducing uncertainties (Kokhanovsky, 2019). Other considerations such as fuel type and emission sources need to be address as smoke plumes largely differ in composition and thus may exhibit different absorption/scattering efficiencies. This study will focus

mainly on oil fire smoke plumes which are far less debated in scientific studies as opposed to biomass burning smoke.

The most abundant source of atmospheric data on oil smoke plumes was gathered from the Kuwait oil fires in 1991. An estimated 700 oil wells were set on fire while smoke plumes engulfed large areas of the Gulf region (Cahalan, 1992). The amount of burned oil was estimated between 1.2 and 7.5 million barrels per day (Sadiq and McCain, 1993). Satellite images of visible smoke where first acquired on the 9[th] of February spanning until November when the last fires were extinguished



(Draxler et al., 1994; Limaye et al., 1992). Several international research teams conducted extensive field campaigns concentrating their efforts on atmospheric, environmental and health related issues focusing on the potential impacts on global climate (Sadiq and McCain, 1993; Husain, Tahir, 1995; World Meteorological Organization, 1993). Studies on health effects related to the Kuwait oil fires suggest that smoke exposure lead to acute respiratory illnesses with some suspecting long-term effects (Etzel and Ashley, 1994; Brain et al., 1998; Smith, 2002; Lange et al., 2002; Kelsall, 2004; Heller, 2011;

Barth et al., 2016). Valuable atmospheric data was also collected from smaller events such as oil depot fires, most notably the Bouncefield incident on 11 December 2005. A number of explosions led to a large fire engulfing 20 storage tanks until the 15[th] of December. The fire burned 58000 tons of fuel while injecting a large smoke plume above the boundary layer at 3000 m (Vautard et al., 2007; Health and Safety Executive and Buncefield Major Incident Investigation Board (Great Britain), 2008). An initial report on air quality concluded that the smoke plume remained aloft over cold and stable

atmospheric layers, thus reducing the potential impacts at ground level (Targa et al., 2006). Health studies related to the event concluded no long-term impacts on people exposed to the smoke however acute respiratory symptoms were reported (Hoek et al., 2007; Morgan et al., 2008).

Oil fires and biomass burnings (BB) smoke plumes significantly differ by OC/BC ratios. Studies show values ranging from 3 to 20 for BB (Andreae and Merlet, 2001; Ichoku et al., 2012; Konovalov et al., 2018; Andreae, 2019; Akagi

et al., 2011) largely dependent on fuel types. This ratio also relates to higher single scattering albedo (SSA) values of 0.7-0.96 for light grey visible smoke plumes (Radke et al., 1991; Dubovik et al., 1998; Eck et al., 1998; Leahy et al., 2007; Pokhrel et al., 2016). However OC/BC ratios are much lower for oil fires: 0.83 – 1.05 (Laursen et al., 1992; Daum et al., 1993; Ferek et al., 1992) also depending on fuel types and have low SSA values: 0.3 – 0.65 (Johnson et al., 1991; Hobbs and Radke, 1992) (Mather et al., 2007; Mikhailov et al., 2006; Gullett et al., 2017) for dark black visible smoke plumes. Studies

also report large BC content in smoke plumes with PM  (particulate matter) ranging from 46 - 50% for Kuwait pool fires (Hobbs and Radke, 1992; Laursen et al., 1992; Stevens et al., 1993) to 50 – 75% for the Bouncefield plume (Mather et al., 2007; Targa et al., 2006) and 75 – 82% for plumes generated by burning oil on the ocean (Ross et al., 1996; Gullett et al., 2016, 2017). These findings suggest that oil smoke plumes heavily absorb light in visible wavelengths and thus require adequate adjustments to any models used for retrieving optical and microphysical properties via remote sensing techniques.

**1.1 Objectives**

One objective of this study is to highlight the importance of satellite remote sensing techniques in identifying these types of events. As opposed to ground-based data, satellite data offers access to remote areas all over the globe which would otherwise be very difficult to achieve. As seen in Table 1, oil installations may be situated in desert areas, at sea or in secluded locations far away from air quality monitoring stations or AERONET (AErosol RObotic NETwork) sites (Holben

et al., 1998). In addition to this advantage, a synergistic approach using different types of satellite instruments can offer three-dimensional space coverage. While in situ, ground stations and modelling tools are viable options for smoke plume research these methods have limitations, in areas prone to armed conflicts or posing high health risks. Out of the



aforementioned events only the event in Kiev was analysed by different techniques, as seen in local AERONET data (section
3). It goes without saying that retrieving optical and microphysical properties of petrochemical burnings may be challenging
in most cases even with this approach. This study will focus on the use of MODIS and CALIPSO aerosol products regarding
these types of aerosol while also highlighting their limitations. By using data from both MODIS instruments on-board Terra
and Aqua satellites we addressed the temporal evolution of the smoke plume while assessing Lidar specific properties and
plume elevation using CALIPSO data. The low number of studies on petrochemical smoke plumes, especially in the last
decade, further encourages us to address these issues. While biomass burning and industrial haze are abundantly discussed in
scientific literature, the same cannot be said for petrochemical smoke plumes resulting from major technological accidents.
To our knowledge, we have not identified any similar studies focused specifically on retrieving aerosol properties from
major petrochemical accidents by using synergistic satellite techniques.

**1.2 Event synopsis**

This section summarizes a collection of events ranging from 2008 to 2019 that were successfully identified by
satellite remote sensing techniques. Table 1 also provides the coordinates and the number of MODIS observation for each of
the events covered in this study.

Events similar in nature to the Kuwait oil fires took place in Northern Iraq as oil fields at Qayyara and Najma were
intentionally set ablaze by Islamic State of Iraq and Syria (ISIS) militants in an attempt to deter coalition air strikes. The first
fires were detected east of Baiji in early January 2016. Other oil wells burned intermittently form May to June close to
Mosul and Kirkuk. The bulk of smoke plumes were observed largely between June and November, however smoke plumes
were continuously detected from Qayyara oil fields for a total of 225 days ranging from 13[th] of June 2016 to the 27[th] of
March 2017. As a result of these event an estimated 1.33 million barrels of oil were burned (Bulmer, 2018; Tichý and
Eichler, 2018). Residents south of the Qayyara oil fields were exposed for a number of 103 days to smoke plumes. Short-
term health effects were reported especially for patients with pre-existing respiratory conditions (Bulmer, 2018).

The Gulf of Sidraa has seen extensive episodes of smoke plumes as oil terminals at As Sidr and Ra's Lanuf, Libya
have been repeatedly set on fire over the course of a decade. These events have been capture by MODIS sensors through the
last decade all the way since 2008. All events were characterized by dark plumes suggesting high contents of BC. On august
19, 2008 a tank fire erupted in Fiba tank farm at Ra's Lanuf after workers failed a maintenance operation (Piafom, 2018; The
Telegraph, 2011). The fire lasted 9 days in which smoke plumes could be detected from the 19[th] to the 22[nd]. In March 2011
the terminal was struck by air artillery in the battle of Ra's Lanuf as the country was engaged in civil war (BBC, 2011; The
Christian Science Monitor, 2011; The Guardian, 2011). Smoke plumes were visible on the 12[th] and 14[th]. In December 2014
the tank farm at As Sidr oil terminal was struck by rockets as rebel fought to seize control of the city port. Seven storage
tanks where engulfed in flames burning 1.8 million barrels of crude oil (Reuters, 2014; BBC, 2014; Al Jazeera, 2014).
Smoke plumes covered large areas of the Gulf between the 26[th] and 30[th] and could be seen as far east as Timimi and Crete.
In January 2016 both tank farms at As Sidr and Ra's Lanuf were struck by Islamic State militants. On January the 5[th] As Sidr





suffered five tank fires while two tanks were hit at Ra's Lanuf amounting to 850,000 barrels of oil. A week later IS militants struck oil infrastructure connecting Ra's Lanuf terminal to other installations in the area (Tichý & Eichler, 2018; Tichý, 2019). A second attack at Ra's Lanuf tank farm was conducted on the 21st (Business Insider, 2016). Throughout the month, extremely dense smoke plumes could be seen in the region from the 5th all the way to the 23rd. The most recent incident at

Ra's Lanuf took place in June 2018 when rival armed groups clashed. The fire which started on the 14th was contained at two storage tanks before it was extinguished several days later (Reuters, 2018). Visible smoke plumes were detected on the 17th and 18th (Bellingcat, 2018).

Tab. 1 Major industrial events leading to observable smoke plumes seen in MODIS RGB images

| Location | Date | Coordinates | Cause of event | Type of installation | Number of observations (days) | References |
|---|---|---|---|---|---|---|
| 1. Qayyara, Iraq | 13.06.2016 | 35.83 N ; 43.21 E | armed conflict | oil wells | 225 | (Tichý and Eichler, 2018) |
| 2. Omidieh, Iran | 06.05.2019 | 30.84 N ; 49.65 E | human error | oil pipeline | 1 | (Financial Tribune, 2019) |
| 3. Haradh, Saudi Arabia | | 24.05 N ; 49.20 E | | | 15 | |
| 4. Hawiyah, Saudi Arabia | | 24.80 N ; 49.35 E | | | 15 | |
| 5. Uthmaniyah, Saudi Arabia | 14.09.2019 | 25.18 N ; 49.31 E | armed conflict | oil processing | 15 | (Khan and Zhaoying, 2020) (Reuters, 2019) |
| 6. Shedgum, Saudi Arabia | | 25.64 N ; 49.39 E | | | 15 | (New York Times, 2019) |
| 7. Buqayq, Saudi Arabia | | 25.92 N ; 49.68 E | | | 15 | |
| 8. Caspian Sea, Azerbaijan | 04.12.2015 | 40.20 N ; 51.06 E | extreme weather | oil and gas platform | 2 | (Necci et al., 2019) |
| 9. Gulf of Mexico, USA | 20.04.2010 | 28.44 N ; 88.21 W | equipment failure | drilling rig | 2 | (Gullett et al., 2016) |
| 10. East China Sea, China | 06.01.2018 | 28.37 N ; 126.08 E | human error | oil tanker | 2 | (Li et al., 2019) (Qiao et al., 2019) |
| 11. Houston Texas, USA | 17.03.2019 | 29.43 N ; 95.05 E | equipment failure | storage tanks | 2 | (An Han et al., 2020) |
| 12. Jaipur, India | 29.10.2009 | 26.77 N ; 75.83 E | Human error | storage tanks | 8 | (Vasanth et al., 2014) |
| 13. Sendai, Japan | 11.03.2011 | 38.27 N ; 141.03 E | earthquake, tsunami | storage tanks | 2 | (Krausmann and Cruz, 2013) |
| 14. Vasylkiv, Ukraine | 08.06.2015 | 50.16 N ; 30.32 E | sabotage | storage tanks | 2 | (Kovalets et al., 2017) (Reuters, 2015) |
| 15. Ra's Lanuf, Libya | 19.08.2008 | 30.45 N ; 18.49 E | human error | storage tanks | 4 | (The Telegraph, 2011) |
| 16. Ra's Lanuf, Libya | 11.03.2011 | 30.45 N ; 18.49 E | armed conflict | storage tanks | 2 | (BBC, 2011) |
| 17. As Sidr, Libya | 26.12.2014 | 30.60 N ; 18.28 E | armed conflict | storage tanks | 4 | (BBC, 2014) |
| 18. Ra's Lanuf, Libya | 05.01.2016 | 30.45 N ; 18.49 E | armed conflict | storage tanks | 3 | |
| 19. As Sidr, Libya | | 30.60 N ; 18.28 E | | | 3 | |
| 20. Surt disrtric, Libya | 14.01.2016 | 30.02 N ; 18.50 E | armed conflict | oil pipeline | 1 | (Tichý and Eichler, 2018) |
| 21. Ra's Lanuf, Libya | 21.01.2016 | 30.45 N ; 18.49 E | armed conflict | storage tanks | 3 | (Tichý, 2019) |
| 22. Ajdaviya district, Libya | 01.02.2016 | 29.68 N ; 20.54 E | armed conflict | oil pipeline | 1 | |
| 23. Ra's Lanuf, Libya | 14.06.2018 | 30.45 N ; 18.49 E | armed conflict | storage tanks | 2 | (Reuters, 2018) |
| 24. Puebla, Mexico | 19.12.2010 | 18.96 N ; 98.45 W | illegal tapings | oil pipeline | 1 | (Biezma et al., 2020) |
| 25. Escravos, Nigeria | 03.01.2018 | 5.45 N ; 5.35 E | bush fire | oil pipeline | 2 | (Bloomberg, 2018) |
| 26. Puerto Sandino, Nicaragua | 17.08.2016 | 12.18 N ; 86.75 W | unknown | storage tanks | 2 | (Ahmadi et al., 2020) |
| 27. Gulf of Oman | 13.06.2019 | 25.39 N ; 57.38 E | armed conflict | oil tanker | 1 | (BBC, 2019) |
| 28. Catano, Puerto Rico | 23.10.2009 | 18.41 N ; 66.13 W | human error | storage tanks | 2 | (Vasanth et al., 2014) |
| 29. Punto Fijo, Venezuela | 25.08.2012 | 11.74 N ; 70.18 W | equipment failure | storage tanks | 1 | (Schmidt et al., 2016) |
| 30. Butcher Island, India | 07.10.2017 | 18.95 N ; 72.90 E | lightning strike | storage tank | 2 | (The Indian Express, |






## 2 Methods and techniques

### 2.1 MODIS aerosol data


The MODerate resolution Imaging Spectroradiometer (MODIS) is a passive remote sensing instrument onboard NASA's Earth Observing Satellites (EOS). The instrument has been collecting climate related data, including aerosol products, since 2000 from Terra and 2002 from Aqua satellite platforms. To achieve a vast catalogue of products MODIS uses its wide spectral range, 36 channels between 0.41 μm and 14.5 μm, fine spatial resolution (250 to 1000 m) broad swath

width (2330 km) and daily temporal resolution.

Herein we will summarize the aerosol retrieval algorithms dark target (DT) over land and ocean (Kaufman et al., 1997; Tanré et al., 1997; Levy et al., 2007a, b, 2009, 2013; Remer et al., 2005, 2008, 2013), and Deep Blue (DB) over land (Hsu et al., 2004, 2006, 2013) with some emphasis on the atmospheric parameters used in the construct of lookup tables (LUT) and aerosol model selection as these properties/assumptions are crucial for proper AOD retrieval in oil smoke events.

The DT land algorithm is used over dark vegetated surfaces with low surface reflectance. DT makes use of the "VIS to 2.1" relationship to distinguish surface contributions to the top-of-atmosphere (TOA) reflectance, as aerosols have a low absorbing and scattering effect in shortwave infrared (2.12 μm) band compared to the visible blue (0.47 μm) and red (0.66 μm) bands. After applying several pixel screening and selection techniques the algorithm chooses specific aerosol models and types based on seasonal and geographical characteristics. From here it determines aerosol related atmospheric

parameters as a function TOA, surface and gas contributions to the apparent reflectance. To achieve AOD inversion these parameters are matched to values within predetermined look up tables (LUT) as an attempt to describe the most likely aerosol conditions. The land algorithm uses five aerosol types composed of two or more models (fine or coarse), each with its specific aerosol optical properties. Primary products retrieved are AOD at 0.55 μm, fine model aerosol type (FMF) and spectral fitting error (ε).

The DT ocean algorithm works in much the same way as the land algorithm although it requires masking sediments and filtering out strong glint areas. It uses the spectral dependencies of six bands, 0.55, 0.65, 0.87, 1.24, 1.63 and 2.12 μm for retrieving surface reflectance while it finds the exact match between the observed and the predetermined LUT reflectance values for the 0.87 μm band. It then attempts the best fit for the remaining bands. The 0.87 μm band is a good indicator of aerosol loading as it is less impacted by water radiance. Preliminary data such as AOD 0.55 μm, reflectance weighting

parameter (η) at 0.55 μm, and aerosol effective radius ($R_{eff}$) are determined before the LUT inversion. DT ocean retrieves the



same products as DT land however it assumes aerosol properties based on a combination of two models, one fine model (four possible modes) and one coarse model (five possible modes). These models are combined by the weighting factor (η) such that the solution yields the lowest fitting error.

Aerosol properties within the LUT such as size distribution parameters, refractive indexes and SSA are crucial for proper aerosol typing and subsequent AOD retrieval. LUT information and model selection is critical for deriving other optical properties, such as Ångström Exponent (AE) which can also be used to describe size distribution.

The DB algorithm was developed to retrieve AOD over arid, semi-arid and urban areas where surface reflectance values are higher than those over dark target regions. The principle behind the algorithm suggest that surface reflectance in these areas show higher values in red and near infrared bands and lower values in the blue band. The algorithm uses reflectance values form 9 bands through each step of the retrieval. After screening and pixel selection surface reflectance values are determined based on three bands (0.412, 0.490 and 0.67 μm) using either a database or a dynamic approach. The approach is selected based on the normalized difference vegetation index (NDVI) and may be a function of region, season, scattering angle and land type. DB matches observed to LUT radiance values using a maximum likelihood method to determine the mixing ratio for a dust and a smoke model. The method retrieves two types of aerosol products: AOD and SSA from the dust model; AOD and AE from the smoke model.

## 2.2 CALIPSO aerosol data

The Cloud Aerosol Lidar with Orthogonal Polarization (CALIOP) onboard the CALIPSO satellite has been observing vertically distributed aerosol and cloud properties since 2006. CALIOP is an elastic backscatter lidar operating at two wavelengths (532 and 1064 nm) equipped with a polarization channel at 532 nm (Hunt et al., 2009; Winker et al., 2009). Calibration is achieved through a molecular normalization technique for night-time measurements at 532 nm which is subsequently the basis for daytime calibrations at both channels (Powell et al., 2009; Kar et al., 2018). The latest CALIOP Version 4 data is significantly improved thanks to the refined calibration algorithms (Getzewich et al., 2018; Kar et al., 2018; Vaughan et al., 2019).

CALIOP data requires several processing sequences, handled by different algorithms, to achieve the desired aerosol and cloud properties (Winker et al., 2009). The first algorithm (selective iterative boundary locator - SIBYL) starts by analysing calibrated level 1 data averaged horizontally (at resolutions from 0.33 km to 80 km) through the use of an adaptive threshold scheme establishing layer boundary limits (Vaughan et al., 2009). The next steps require the use of scene classification algorithms (SCA). Firstly, the cloud-aerosol discrimination (CAD) algorithm uses multidimensional probability density functions (PDFs) to distinguish clouds from aerosol layers (Liu et al., 2005, 2009). The primary inputs from the CAD algorithm is later used for subtyping aerosol species throughout the troposphere and stratosphere (Omar et al., 2009; Kim et al., 2018). Finally, SCA uses the attenuated backscatter and volume depolarization ratios (both layer-integrated) to distinguish between water and ice clouds (Hu et al., 2009; Avery et al., 2020). To extract aerosol properties



(particulate backscatter and extinction coefficients, optical depth) the classified layer data is fed through several hybrid extinction retrieval algorithms (HERA) (Young and Vaughan, 2009; Young et al., 2013, 2018).

Lidar ratios are essential for calculating extinction coefficients, and throughout these sequences of algorithms lidar ratios are selected in one of two ways. For unconstrained retrievals, lidar ratios are selected based on the aerosol subtype classification which is a function of surface type, location, particulate depolarization ratio, integrated attenuated backscatter (Omar et al., 2009; Kim et al., 2018). For each aerosol subtype a lidar ratio is assigned based on AERONET data, direct measurements and theoretical scattering calculations (Omar et al., 2009; Tackett et al., 2018). The second approach, known

as constrained retrievals, is based on measured layer two-way transmittance (Young and Vaughan, 2009; Young et al., 2018). Selection between these two approaches is done based on scene complexity and feature classification (Young and Vaughan, 2009; Young et al., 2018). In most cases aerosol lidar ratios are determined using unconstrained retrievals (e.g., layers in contact with the surface) however constrained solutions are possible in certain situations (Young and Vaughan, 2009; Young et al., 2018).

**2.3 Synergic approach**

    Figure 1 summarizes each step of the analysis starting from a collection of events identified in scientific literature and local news outlets. We chose a 12-year time frame in which both MODIS and CALIPSO retrieved a substantial amount of aerosol properties originating from oil smoke plumes. By using MODIS red, green and blue RGB images we can visually identify each plume (Table 1). The two MODIS sensors onboard Aqua and Terra were used as they possess a number of

advantageous characteristics for plume identification and analysis: daily global coverage, good pixel resolution, algorithm maturity, two retrieval windows, large data record (20 plus years of mission). CALIPSO data was used to compare plume AOD and to fill the gaps in MODIS data such as: plume thickness and elevation, scene classification, aerosol typing (lidar ratio and particulate depolarization ratio). Events identified in literature were selected for analysis based on plume dimensions and retrieval conditions. We selected plumes larger than 500 km$^2$ for statistical relevance as smaller plumes

resulted in a low pixel count. Events were discarded if the atmospheric scene was predominantly cloudy, over 50% cloud coverage. We deemed retrievals successful if they produce AOD pixel values, with some degree of variation (For at least 50% of pixels, AOD should vary resulting in value differences of at least 0.01), overlapping the plume area. We deemed unsuccessful retrieval if no AOD data was retrieved over the plume area (after cloud screening), or in cases where values of AOD were lower than 0.1 and the retrieved pixels were homogeneous (ex: over 90% of plume pixels with a fixed AOD

value of 0.09 as seen in figure 6d). We used both successful and unsuccessful retrievals to highlight the capabilities and limitation of the MODIS sensor. We used both MODIS Aqua and MODIS Terra AOD products, collection 6.1 (MODIS Atmosphere Science Team, 2017a, b), for successful retrievals. If products from both sensors were available for unsuccessful retrievals we selected only the most relevant scene. The algorithm for the AOD products was selected based on surface type (DT over ocean and land) and locations (DB over desert and arid areas) while aerosol properties were analysed only for

successful retrievals. Based on our preliminary findings the DT ocean products were selected for successful retrievals while





DT land and DB land products were used for unsuccessful retrievals. We took advantage of the higher resolution 3 x 3 km$^2$ level 2 AOD products for statistical relevance in successful retrievals over ocean. For unsuccessful retrievals we used the 10 x 10 km$^2$ level 2 AOD products (DB over desert and arid areas) and the 3 x 3 km$^2$ level 2 AOD products over land. The following aerosol properties were used in our analysis for successful retrievals: AOD at 0.55 μm, AE and R$_{eff}$. To highlight

the MODIS AOD product limitations we used AOD at 0.55 μm in scenes with unsuccessful retrievals. For successful retrievals, we developed a plume averaging technique based on plume and background AOD values. Since both the RGB and AOD images show visible distinctions over the plume area, as seen in figures 2a and 2b., we constructed a plume edge based on AOD pixel gradient. The pixels selected to represent the plume edge share a unique AOD value which is different form the neighbouring background pixels by a value of at least 0.03. We averaged the AOD values from the plume area, within

the plume edge, to obtain "plume average AOD". To achieve "plume specific AOD" values we subtracted a local averaged background AOD value from the plume average AOD value. The result should represent the average AOD contribution of the oil smoke aerosols to the total column AOD, as seen in figure 2.c. Local background pixel count was chosen between 3 to 10 times the number of plume pixel count based on the local geography and meteorological conditions. A larger background pixel count is expected in scenes with low cloud coverage; over open water as seen in figures 3a and 3b. A

smaller pixel count is expected in scenes with high cloud coverage over smaller gulf regions as seen in figures 3d and 3e. Results and discussions for both successful and unsuccessful MODIS retrievals are presented in section 3.1.

One advantage of using CALIPSO data is that retrievals are done within 2 minutes following MODIS Aqua, for daytime retrievals, and thus plume and atmospheric conditions are similar for both sensors. Events were selected based on the plume cross section dimension. We used total attenuated backscatter (532 nm) values at single shot resolution to

determine the extent of the plume cross section. Similar to MODIS data, we could identify a plume signature as backscatter values in the cross section were larger than background levels. We chose plume cross sections larger than 5 km as level 2 data is averaged starting from this resolution. We used level 1B profile data (532 nm), standard version 4.10, both day and night time retrievals (Winker, David, 2016), to identify the plume signature. For daytime retrievals we used the Aqua MODIS RGB image prior to the CALIPSO retrieval for visual confirmation of the smoke plumes. For nigh time CALIPSO

retrievals we used two consecutive daytime MODIS RGB images for an indirect "visual confirmation" of the smoke plume. In these cases we used a MODIS RGB image the day prior to the CALIPSO retrieval and the first MODIS RGB image directly after the CALIPSO retrieval to assess the plume spatial continuity. We used Level 2 data (532 and 1064 nm), 5 km Aerosol Profile, standard version 4.20, both day and night time retrievals (Winker, David, 2018b) for statistical analysis of aerosol optical properties and comparison to MODIS AOD products. For aerosol typing and feature classification we used

level 2 data of the Vertical Feature Mask product, standard version 4.20 (Winker, David, 2018c). We used the 5 km Aerosol Layer product, standard version 4.20, both day and night time retrievals (Winker, David, 2018a) to assess the lidar ratio (532 and 1064 nm) if the plume was identified as a distinct aerosol feature. Focusing on the plume bins we used the extinction coefficient to backscatter ratio to determine the plume averaged lidar ratios if the plume was not identified as a distinct aerosol feature.



We used a quality filtering procedure similar to the one described by Tackett et al., 2018. For cloud-free data we used only Aerosol Profiles with a cloud-aerosol-discrimination (CAD) score of - 100 < CAD score < - 20. We discarded aerosol profiles in all range bins directly below any type of clouds as these profiles may be affected. Smoke plume profiles were discarded if the average plume height exceeded 4 km (above mean surface level) to prevent aerosols in contact with ice clouds and subsequent misclassifications and cirrus contamination. For the extinction filtering procedure, QC flag values not

equal to 0, 1, 16, or 18 were discarded as low-confidence retrievals. Extinction samples where extinction uncertainty is 99.9 km$^{-1}$ and retrievals in bins directly below these samples were rejected as these values may also be affected.

      We selected the following level 2 aerosol data in the analysis: Particulate backscatter coefficients, extinction coefficient (532 and 1064 nm) and particulate depolarization ratio (532 nm). Based on the particulate total backscatter coefficient (532 nm) we defined the plume cross section as in each range bin, the plume values were at least 2 times higher

than background values. Based on this parameter we also defined the plume thickness and elevation. We defined the background region as the same number of bins and of the same height as plume bins, situated either upwards or downwards of the plume location (depending on plume trajectory), following the overpass trajectory. The extinction values (532 and 1064) in plume bins were used to integrate the AOD for each plume profile. The resulting AOD values were averaged to obtain a plume mean AOD value. For both plume cross section and background area we averaged the aerosol optical

properties (particulate backscatter, extinction and depolarization ratios). To assess the possibility of aerosol typing we averaged the plume lidar ratios and calculated the Ångström exponent using the plume mean AOD values (532/1064 nm). Averaged data was compared between plume cross sections and local background values to assess CALIPSO feature detection and aerosol typing algorithms. We compared our data between events to determine how lidar ratio solutions affect aerosol properties estimates. Based on particulate depolarization ratios and lidar ratios we assessed the possibility of oil

smoke aerosol typing either as a distinct class or as part of the predetermined CALIPSO aerosol types.

      Since both MODIS and CALIPSO identified plume specific AOD and AE values we compared our results to determine each methods strengths and weaknesses. We also identified one AERONET event which was also used to compare plume values. We compared our results to other studies done on oil smoke plumes in which the same aerosol properties were determined by means of ground based or flight measurements. Our conclusions reflect the nature of these

smoke plumes and the implications they have on current satellite retrieval capabilities.



---

**Event identified over a 12 year period (2008-20019) based on scientific literature and news outlets**

**MODIS**  |  MODIS RGB for plume visual confirmation  |  **CALIPSO**

**AERONET**

**Event selection criteria**

a) Plumes larger than 500 km²
b) Cloud coverage less than 50%
c) Use data if retrieval is deemed successful*
d) Use data if retrieval is deemed unsuccessful*
  if a) & b) are met
* successful if pixel AOD ≥ 0 and Stdev ≥ 0.001
* unsuccessful if pixel AOD = no data or Stdev ≤ 0.001

**Sensor data selection**

(MODIS collection 6.1)

Successful retrievals – Aqua & Terra
Unsuccessful retrievals – Aqua or Terra

**Retrieval algorithm selection**

DT ocean for successful retrievals
DT land & DB land for unsuccessful retrievals

**AOD product selection**

(Level 2 data)

3 x 3 km for DT ocean and land
10 x 10 km for DB land

**Selected aerosol properties**

AOD at 55 μm, AE and R$_{eff}$ for successful retrievals
AOD at 55 μm for unsuccessful retrievals

**Plume averaging criteria**

- AOD pixels overlapping plumes (AOD & RGB)
- Defining plume edge:
Gradient of plume pixels with unique AOD value ≥ 0.03 + AOD of neighboring background pixels
- Obtaining mean plume values by averaging plume AOD pixels not exceeding the plume edge
- Defining the local background:
Mean AOD value outside the plume edge representing the local scene. 10x< Background pixel count >3x plume pixel count, depending on geography and local atmospheric scene.
- Obtaining plume specific AOD values by subtracting plume pixel AOD from the mean background AOD.
- Obtaining mean plume AE and R$_{eff}$ by averaging values within the previously defined plume edge

**Compare results between individual events**

• Mean plume vs mean background values
• Discuss conditions leading to successful or unsuccessful retrievals

**AEROENT data (V3)**

• AOD
• AE
• Inversion products

**Event selection criteria**

a) CALIPSO overpass results in plume cross section
b) No cloud coverage
c) Plume cross section > 5 km
d) Total attenuated backscatter (532 nm) values in plume cross section > larger than background values

**CALIPSO data selection**

a) Level 1B profile data, standard version 4.10, day and night:
• Total attenuated backscatter (532 nm) for identifying the plume signature.
b) Level 2 data, 5 km aerosol profile, standard version 4.20, day and night:
• Particulate backscatter coefficients (532 & 1064) for plume cross section, thickness and elevation
• Extinction coefficients (532 & 1064 nm) for AOD integration
• Particulate depolarization ratio (532) for aerosol typing
c) Level 2 data, Vertical Feature Mask, standard version 4.20, day and night, for feature classification, quality assurance and aerosol typing
d) Level 2 data, 5 km Aerosol Layer, standard version 4.20, day and night:
• Feature lidar ratio (532 & 1064 nm) for aerosol typing if the plume is identified as a distinct feature

**Quality filtering procedure**

• -100 < CAD score < -20
• Aerosol profiles discarded in all range bins directly below cloud features
• Aerosol profiles discarded in range bins above 4 km to prevent cirrus misclassification
• Extinction QC flag values not equal to 0, 1, 16, or 18 discarded as low-confidence retrievals
• Extinction uncertainty of 99.9 km⁻¹ and retrievals in bins directly below these samples were rejected

**Plume averaging criteria**

(Level 2 data)

• Plume cross section defined as particulate total backscatter coefficient (532 nm) values ≥ 2x background values in each range bin.
• Background region defined as the same number of bins and of the same height as plume bins, situated either upwards or downwards of the plume location (depending on plume trajectory), following the overpass trajectory.
• Mean plume aerosol properties (particulate backscatter coefficients, extinction coefficient and particulate depolarization ratio) within the plume cross section

**Compute aditional aerosol properties**

• Mean plume AOD (532 & 1064 nm) from integrated extinction values in plume bins
• Mean AE (532/1064 nm) from mean plume AOD
• Mean Lidar ratios (532 & 1064 nm) from backscatter to extinction ratios in plume bins

**Compare results between individual events**

• Mean plume values vs mean background values
• Plume feature detection
• Implication for aerosol typing (Lidar ratio vs particulate depolarization ratio vs CALIPSO aerosol typing)

**Compare results between MODIS, CALIPSO, AERONET vs other studies**

• Compare results between MODIS, CALIPSO, AERONET vs other studies (ground based or flight measurements)
• Conclusions and implications



Fig. 1 Flowchart of the plume analysis method

# 3 Results and discussions

## 3.1 MODIS data

Figure 2. Visual representation of the analysis method for MODIS data: (a) - plume captured in true color; (b) - AOD retrieval over the plume area and background (Gulf of Sidra); (c) - AOD retrieval as a result of plume minus background values; (d) – Angstrom exponent for plume and background area; (e) – Effective radius for plume and background area. The red coloured "x" indicates the event origin
(Satellite imagery from the NASA Worldview application, https://worldview.earthdata.nasa.gov).

We selected a successful retrieval to better describe the method used for our analysis. Figure 2 shows the case at Ra's Lanuf and As Sidr tanks farms which caught fire on the 5[th] of January 2016 and burned throughout the 6[th] and 7[th]. The retrieval in the images was taken on the 6[th] by MODIS Aqua at 12:05 UTC. Figure 2a represents a true colour composite
image showing the smoke plumes emerging from both sites and travelling E-NE over the Gulf of Sidra. Judging by this image alone we can only distinguish parts of the smoke plume which appear to be less dispersed and thus present a smaller



mixing ratio with the local background aerosols. In this study, we focused our attention on the plume areas where heavy concentrations of aerosol are present while discarding retrievals done at the edges of the plume where background aerosol may have a large influence on the values retrieved. Thus figure 2b was constructed based on the AOD (0.55 µm) retrieval

and a plume edge selection technique. To determine the plume edge we constructed isolines of AOD values from the retrieval pixels. The 3×3 km product is better suited for determining the AOD gradients and thus was selected over the standard 10×10 km product. Figure 2b shows higher values of AOD in the selected plume area as opposed to the local background levels. At the time of the retrieval, we can observe two distinct plumes of smoke, a thin plume originating from the As Sidr and the main plume (within the black contour, Fig. 2b) originating from Ra's Lanuf. Since the As Sidr plume did

not meet the selection criteria the analysis is made for the main Ra's Lanuf plume. To further discriminate between plume and background AOD values we averaged all non-plume pixel values, over water, within the gulf region without considering the pixels of the plume. Then, the averaged AOD values were subtracted from each pixel of the plume AOD values to determine the overall plume contribution. Consequently, Figure 2c illustrates the plume specific AOD gradient. Figures 2d and 2e show the AE (0.55/0.86 µm) and the $R_{eff}$ (µm) which were selected for aerosol typing. For AE and $R_{eff}$ we used the

same edge selection technique as described above without the background subtraction. The AE is shown to have very low values indicating a dominant coarse mode which is further evidenced by the large $R_{eff}$ chosen from the LUT. It is also evident from both figures that both plumes extend further from the edge selection. Mean and standard deviation values are presented in table 2 and 3 based on both Aqua and Terra MODIS retrievals.





Figure 3. Plume specific corrected AOD for: (a) SOCAR's Platform No. 10, Caspian Sea, 08.12.2015 (b) Deepwater Horizon, Gulf of Mexico, 21.04.2010 c) Sendai, Japan, 11.03.2011 d) Ra's Lanuf, Libya, 21.01.2016, e) As Sidr, Libya, 28.12.2014, f) As Sidr, Libya, 29.12.2014, g) As Sidr, Libya, 30.12.2014, h) Puerto Sandino, Nicaragua, 19.08.2016 and i) Escravos, Nigeria, 04.01.2018. The red coloured "x" indicates the event origin (Satellite imagery from the NASA Worldview application, https://worldview.earthdata.nasa.gov).

Table 2. Mean and standard deviation values of aerosol properties (AOD, AE, $R_{eff}$) based on MODIS Terra successful retrievals

| Event date | AOD plume | AOD background | AOD (plume | AE plume | AE background | $R_{eff}$ plume | $R_{eff}$ background |
|---|---|---|---|---|---|---|---|



| | | specific) | | | | |
|---|---|---|---|---|---|---|
| 08.12.2015 | 0.19; 0.04 | 0.06; 0.01 | 0.13; 0.04 | 1.25; 0.18 | 1.59; 0.44 | 0.47; 0.06 | 0.29; 0.23 |
| 21.04.2010 | 0.25; 0.03 | 0.20; 0.02 | 0.05; 0.03 | 0.34; 0.25 | 1.17; 0.30 | 0.61; 0.14 | 0.26; 0.05 |
| 11.03.2011 | 0.29; 0.05 | 0.13; 0.05 | 0.16; 0.05 | 0.43; 0.30 | 1.64; 0.61 | 0.65; 0.19 | 0.22; 0.10 |
| 06.01.2016 | 0.12; 0.02 | 0.02; 0.005 | 0.10; 0.02 | -0.18; 0.002 | 0.45; 0.38 | 1.45; 0.02 | 0.51; 0.16 |
| 21.01.2016 | 0.21; 0.03 | 0.07; 0.02 | 0.14; 0.03 | -0.13; 0.09 | 1.20; 0.33 | 1.34; 0.29 | 0.36; 0.12 |
| 28.12.2014 | 0.22; 0.05 | 0.07; 0.02 | 0.15; 0.05 | -0.07; 0.15 | 0.68; 0.33 | 1.19; 0.22 | 0.49; 0.14 |
| 29.12.2014 | 0.13; 0.02 | 0.05; 0.004 | 0.08; 0.02 | -0.03; 0.06 | 0.52; 0.12 | 1.03; 0.16 | 0.79; 0.10 |
| 30.12.2014 | 0.18; 0.03 | 0.15; 0.08 | 0.03; 0.07 | -0.11; 0.10 | 0.08;0.14 | 1.48; 0.31 | 0.80; 0.15 |
| 19.08.2016 | 0.24; 0.03 | 0.19; 0.04 | 0.05; 0.03 | 0.06; 0.16 | 0.41; 0.20 | 0.87; 0.12 | 0.61; 0.10 |

Table 3. Mean and standard deviation values of aerosol properties (AOD, AE, Reff) based on MODIS Aqua successful retrievals

| Event date | AOD plume | AOD background | AOD (plume specific) | AE plume | AE background | $R_{eff}$ plume | $R_{eff}$ background |
|---|---|---|---|---|---|---|---|
| 08.12.2015 | 0.14; 0.03 | 0.03; 0.01 | 0.11; 0.03 | 0.96; 0.35 | 1.28; 0.34 | 0.29; 0.06 | 0.28; 0.14 |
| 21.04.2010 | 0.23; 0.03 | 0.16; 0.02 | 0.07; 0.03 | 0.74; 0.27 | 1.41; 0.24 | 0.38; 0.13 | 0.26; 0.06 |
| 11.03.2011 | 0.24; 0.04 | 0.14; 0.03 | 0.10; 0.04 | 0.50; 0.19 | 0.85; 0.21 | 0.57; 0.13 | 0.36; 0.09 |
| 06.01.2016 | 0.21; 0.05 | 0.08; 0.04 | 0.13; 0.05 | -0.14; 0.08 | 0.38; 0.39 | 1.64; 0.37 | 0.68; 0.22 |
| 21.01.2016 | 0.15; 0.02 | 0.05; 0.01 | 0.10; 0.02 | -0.15; 0.07 | 0.79; 0.21 | 1.38; 0.16 | 0.32; 0.07 |
| 28.12.2014 | 0.11; 0.02 | 0.05; 0.01 | 0.06; 0.02 | -0.13; 0.15 | 0.01; 0.18 | 1.44; 0.05 | 1.04; 0.16 |
| 29.12.2014 | 0.15; 0.05 | 0.07; 0.03 | 0.08; 0.05 | -0.06; 0.15 | 0.52; 0.30 | 1.73; 0.45 | 0.70; 0.17 |
| 30.12.2014 | 0.13; 0.04 | 0.16; 0.04 | -0.03; 0.03 | -0.11; 0.11 | 0.09; 0.14 | 1.37; 0.12 | 0.89; 0.13 |
| 19.08.2016 | 0.09; 0.01 | 0.12; 0.03 | -0.03; 0.01 | -0.01; 0.19 | 0.31; 0.34 | 1.17; 1.29 | 0.71; 0.20 |
| 04.01.2018 | 0.75; 0.09 | 0.79; 0.07 | -0.04; 0.09 | 0.78; 0.29 | 0.67; 0.23 | 0.58; 0.14 | 0.54; 0.14 |

Next, we examined the smoke plume from SOCAR's Platform No.10 fire in the Caspian Sea as an "atypical" event

based on the fuel type and plume albedo. According to the available information (Business-humanrights, n.d.), Platform

No.10 was operating 24 oil wells and 4 gas wells of which all 4 gas wells were involved in the fire. The resulting smoke

plume can be seen from MODIS true colour images exhibiting a light grey, whitish colour. The plume albedo is more similar

to biomass smoke and less similar to black smoke plumes as seen from the events presented in this study. Thus, a correct

LUT construction and matching of the observed atmospheric properties should take into account the plume's SSA. In figure

3a we observe the plume's specific AOD and AOD gradient computed from MODIS Aqua retrieval. As seen in Tables 2 and

3, mean AOD plume specific values fall between two and three times higher than local background values. Within a three-

hour window between the two retrievals (Terra vs Aqua) we can observe a slight drop in AOD values as the plume is being

dispersed, however these small changes may also be attributed to other factors such as the satellite sensor calibration and

retrieval geometry. As opposed to Platform No.10 fire, the Deepwater Horizon fire (Gulf of Mexico) was mainly fuelled by

oil. This is evident in the plume albedo from MODIS true colour images. The gradient in figure 3b suggests higher AOD

values closer to the platform location where high concentrations of BC are expected. The plume shape is a result of a change

in wind speed and direction that took place within the 3-hour retrieval window between Terra and Aqua. AOD values

remained similar for both MODIS sensors while plume specific values where slightly above background levels. The JX



Nippon fire produced AOD values two times higher than the background values, within the first hours of the event, after
which values declined slowly at the time of Aqua overpass. Figure 3c shows a plume specific AOD values ranging from 0.06
to 0.23. Figure 3c shows AOD values as high as 0.24 over the average AOD background level for the plume originating at
Ra's Lanuf. During this event, an increase in AOD levels between the two retrievals was observed as the fire spread to
several oil tanks. The AOD gradient, in figure 3c, shows the largest values at the centre of the plume where aerosol mixing is
expected to be lower. Two weeks later the same tank farm injected thick plumes, figure 3d. over the Surt district in Libya.
The plume section over the Gulf recorded AOD values twice as high as the background level however the net contribution
amounted, on average, to a value of 0.10. Plumes from As Sidr are visible in Figs. 3e, 3f, and 3g. This event was captured in
multiple days while the fire engulfed several oil tanks and subsequently injecting higher amounts of aerosols in the region.
Depending on the local background levels, average plume specific values ranged from -0.03 to 0.15. Negative values can be
explained by the presence of dust and marine aerosols in the atmospheric background. This is especially evident in figure 3g
where high background levels were registered in the Gulf of Sidra while lower levels were seen off the shores of At Tamimi,
600 km NE of As Sidr. The Puerto Sandino fire produced a thinner layer of smoke which was evident in the AOD retrieval
with values close to the background level. As seen in figure. 3h the smoke plume is largely dispersed and thus exhibiting
lower levels of AOD as opposed to 3 hours earlier (see table 2). One would expect higher values of AOD over the plume
area however in cases where background levels are already low, 0.12, a thin layer of smoke may result in lower TOA
reflectance and AOD values. In the case of the Escravos fire, local background AOD levels were influenced by heavy dust
and biomass smoke with average values of 0.79. Results show slightly negative values for plume specific AOD which could
also be influenced by the optical properties of black smoke, as these types of aerosol are highly absorbent in visible
wavelengths and could potentially lower TOA reflectance and AOD values.


Figure 4. Ångström exponent retrieved: (a) (a) SOCAR's Platform No. 10, Caspian Sea, 08.12.2015 (b) Deepwater Horizon, Gulf of Mexico, 21.04.2010 c) Sendai, Japan, 11.03.2011 d) Ra's Lanuf, Libya, 21.01.2016, e) As Sidr, Libya, 28.12.2014, f) As Sidr, Libya, 29.12.2014, g) As Sidr, Libya, 30.12.2014, h) Puerto Sandino, Nicaragua, 19.08.2016 and i) Escravos, Nigeria, 04.01.2018. The isolines indicate the boundaries of the smoke plume. The red coloured "x" indicates the event origin (Satellite imagery from the NASA Worldview application, https://worldview.earthdata.nasa.gov).

We computed the Ångström exponent (0.55/0.86 μm) for plume and background values as previously described. The outlier case from SOCAR's Platform No. 10 fire shows a slight difference in AE plume values with respect to the background aerosols. As seen in table 2 and table 3, plume values slightly decreased between the two sensor retrievals





however these values suggest the presence of fine mode dominant aerosol, similar to biomass smoke and marine aerosols in the local background. The Deepwater Horizon fire produced larger particles judging by the AE, 0.34 and 0.74 for

MODIS/Terra and MODIS/Aqua respectively. As discussed in the previous section, this could be a result of the different fuel type, mostly oil and less natural gas. Plume values close to 0 were retrieved near the event while average values registered two to three times lower than the local background. The Sendai plume resulted in AE levels of 0.43 and 0.50 for MODIS/Terra and MODIS/Aqua respectively also two to three times lower than the local background. In the case of the Libyan events all AE plume values were below 0 suggesting a coarse dominant scene. Figure 4g shows low AE values also

identified further from the plume edge showing the spatial extent of these types of aerosols. The Gulf of Sidra is situated in one of the main pathways of long range transported dust (Kallos et al., 2007) thus affecting AE local background values, as seen in table 2 and table 3. While AE plume values are generally low, these extremely low values may not be primarily a direct result of particle size distribution. MODIS uses spectral reflectance relations to determine AOD and subsequently AE levels. While other types of aerosols have a varying spectral reflectance signature, heavy concentrated black carbon exhibit a

flat and linear signature that result in low spectral reflectance values (Johnson et al., 1991; King, 1992; Pilewskie and Valero, 1992; Soulen et al., 2000). In the case of Puerto Sandino, AE values are also close to 0 while background values fluctuate between 0.31 and 0.41. These low background values may be a result of a clean atmospheric background. Escravos was the only event where AE plume values were larger than background values. At the time of the retrieval, a heavy dust intrusion was spotted over the Gulf of Guinea, thus the higher AE can be attributed to mixing of smoke and dust particles in

the local atmospheric scene. Except for the Gunashli event, all the plumes exhibit AE values lower than 1 and judging from AE gradients in figure. 4, the lowest values were identified closer to the sites where larger aerosol particles are expected.





Figure 5. Effective radius retrieved from plume areas and local background: (a) SOCAR's Platform No. 10, Caspian Sea, 08.12.2015 (b) Deepwater Horizon, Gulf of Mexico, 21.04.2010 c) Sendai, Japan, 11.03.2011 d) Ra's Lanuf, Libya, 21.01.2016, e) As Sidr, Libya, 28.12.2014, f) As Sidr, Libya, 29.12.2014, g) As Sidr, Libya, 30.12.2014, h) Puerto Sandino, Nicaragua, 19.08.2016 and i) Escravos, Nigeria, 04.01.2018. The isolines indicate the boundaries of the smoke plume. The red coloured "x" indicates the event origin (Satellite imagery from the NASA Worldview application, https://worldview.earthdata.nasa.gov).

To further distinguish between these events and the atmospheric background we selected the effective radius based on MODIS LUT. For the ocean algorithm $R_{eff}$ values range from 0.10 to 2.50 µm with lower values being a good indicator of





fine dominant aerosols whereas higher values of coarse dominant aerosol type. Figure 5 shows $R_{eff}$ values for plume areas and the local background. The SOCAR's Platform No. 10 case does not show any significant change in values between the plume area and background levels (figure 5a). As previously discussed, this is due to the specific nature of the event. The low $R_{eff}$ values suggest a fine dominant aerosol type similar to aged biomass smoke or sea salt. Low values are also

identified for the Deepwater Horizon events although Terra MODIS observed values 3 times higher than background levels. Figure 5b shows larger values close to the platform location where larger particles are expected. The Sendai plume (figure 5c) registered consistent values between the two sensors with values over 1 μm close to the event. The events at Ra's Lanuf and As Sidr registered the highest $R_{eff}$ values consistently over 1 μm while, in some cases, values close to the event and within the centre of the plume area reached the maximum 2.50 μm. These large values are consistent with the observed AE

trend observed indicating larger particles and coarse mode dominant aerosol type. Background values for these events fluctuated between 0.32 and 1.04 μm due to regional dust-like aerosols. The Puerto Sandino plume registered higher values from the Aqua sensor however these values widely fluctuated within the plume area and may be attributed to spatial and temporal differences of the two MODIS sensors. For the Escravos plume $R_{eff}$ values were heavily influenced by a presence of biomass smoke and dust aerosols in the region, thus retrieved values mirrored the background levels. Judging by these

background levels the local atmospheric scene was fine mode dominant indicating a heavy biomass smoke contribution over dust-like aerosols. Based on the ocean algorithm, $R_{eff}$ values are grouped within specific intervals in the construction of the LUTs. Thus, higher values are generally reserved for dust heavy atmospheric conditions and less so for heavy smoke.

         After analyzing plume values seen in figures 2 through 5 and tables 2 and 3 we can determine that MODIS algorithm over ocean is able to detect large changes in AE and $R_{eff}$ values. Aerosol load however is only slightly impacted by

the presence of heavy smoke and thus to validate these results, other datasets such as CALIPSO and AERONET data were used.







Figure 6. Retrieval of plume (unsuccessful) and background AOD values: (a) Haradh, Hawiyah, Uthmaniyah, Shedgum, Buqayq, Saudi Arabia 14.09.2019, (b) Qayyara, Iraq, 09.10.2016, (c) Omidieh, Iran, 06.05.2019, (d) As Sidr, Libya 28.12.2014, (e) Jaipur, India, 30.10.2009 (f) San Martín Texmelucan de Labastida of Puebla, Mexico, 19.12.2010 and (g) Vasylkiv, Kyiv Oblast, Ukraine, 09.06.2015. The red coloured "x" indicates the event origin (Satellite imagery from the NASA Worldview application, https://worldview.earthdata.nasa.gov).

Figure 6 shows all unsuccessful retrievals of plume AOD using the land algorithm. Figures 6a - e were constructed using the 10 km Deep Blue product since these events took place in arid and semi-arid regions while Figure 6f and g show



the 3 km Dark Target product. The cases form Saudi Arabia, Iran and Iraq show no values retrieved over the plume areas. This may be a result of the algorithm not finding a proper fit to the LUT. In these cases, radiance, surface and TOA reflectance values are lower as a result of the smoke plumes low transmission and albedo. Thus, predetermined values within the LUT may not have included such "extreme" cases. Figure 6d shows the event at As Sidr also captured by the ocean

algorithm and analysed in the previous section (successful retrievals). In this case we can distinguish the plume from the RGB image over the Gulf of Sidraa while also observing AOD values over land where the smoke plume drifted E-NE towards the island of Crete. However, there seems to be no distinguishable AOD gradient within the plume. A further inspection suggested that all pixels showed values of 0.095 which suggest that the lower radiance values did not match well with pre-existing LUT values. Consequently, the region is classified as "clean atmosphere" and thus, a unique AOD value is

assigned to all the pixels. Conversely, the ocean algorithm retrieved AOD that varied between 0.1 and 0.37. The same situation emerged from the Jaipur event (figure 6e) where plume pixels did not present any AOD gradient with only pixel AOD values of 0.095. Background values are much higher averaging 0.3 while biomass burning in the area recorded values of AOD over 1. It is safe to say that AOD values for black smoke aerosol are not correctly retrieved over land surfaces due to lower atmospheric transmission all values are either incorrectly selected by LUT or are assigned to a "clean atmosphere"

value. Figure 6f and 6g show the unsuccessful retrievals for San Martín Texmelucan de Labastida of Puebla, Mexico and Vasylkiv, Kyiv Oblast, Ukraine. It was expected for both cases that larger values of AOD should be observed over plume pixels with lowest albedo. However, in these regions the land algorithm produces fill values and thus, no AOD value could be retrieved. In figure 6f we can identify higher AOD values west of the black plume area that correspond to local biomass burning events and are successfully retrieved since these atmospheric conditions are included in the LUT. It is also obvious

that at the edge of the black plume, the algorithm retrieved a negative value which leads us to believe that values lower than -0.10 were identified within the plume. This may only be the result of unmatched MODIS observed to LUT values. Since these heavy smoke plumes are the result of extreme scenarios they are rarely observed and may not end up being a subject of research. Thus, we believe there are no cases within the LUT values describing extremely low atmospheric transmission and radiance values, highly absorbent aerosol, low SSA and low reflectance values over a large spectral range including MODIS

bands 1 through 7.

### 3.2 CALIPSO Data

The event at Ra's Lanuf and As Sidr, 6 January 2016, was also captured by CALIPSO lidar measurements as CALIPSO overpass matched a cross section of the plume area. Figure 7a shows this overlap in near-real time as CALIPSO

succeeds Aqua within a 2 minute time frame. Within the 15 km plume cross section we selected a particulate backscatter coefficient profile for reference, figure 7b, and based on this parameter we determine plume elevation and thickness. The average plume thickness was approximately 920 m ranging from 2700 m to 3300 m. The entire plume cross section is presented in figure 8. We observe the main plume from Ra's Lanuf elevated between 2400 m and 4200 m. Figure 8 also





shows the secondary plume from As Sidr, 0.2° north of the main plume situated around 2000 m. Based on CALIPSO

measurements of the main plume, table 4, average particulate backscatter (532 nm) values measured 0.015 km$^{-1}$sr$^{-1}$ while

values at 1064 nm measured 0.17 km$^{-1}$sr$^{-1}$. Average extinction coefficient values at 532 nm were measured at 1.65 km$^{-1}$

while the 1064 nm channel yielded a value of 1.55 km$^{-1}$. This event is an example of an opaque aerosol layer, were the lidar

did not penetrate the plume up to the sea surface over the Gulf of Sidra. Table 6 shows a lidar ratio of 109 sr at 532 nm and

86 sr at 1064 nm. These values are larger than the CALIPSO V4 aerosol subtype values for: elevated smoke 70 ± 16 (532

nm) and 30 ± 18 (1064 nm); Polluted continental/smoke 70 ± 25 (532 nm) and 30 ± 18 (1064 nm) (Kim et al., 2018). The

particulate depolarization ratio for the Ra's Lanuf plume was 0.11 that corresponds to moderately depolarizing smoke (Kim

et al., 2018). Figure 8c shows the CALIPSO feature classification while figure 8b shows the aerosol typing results.

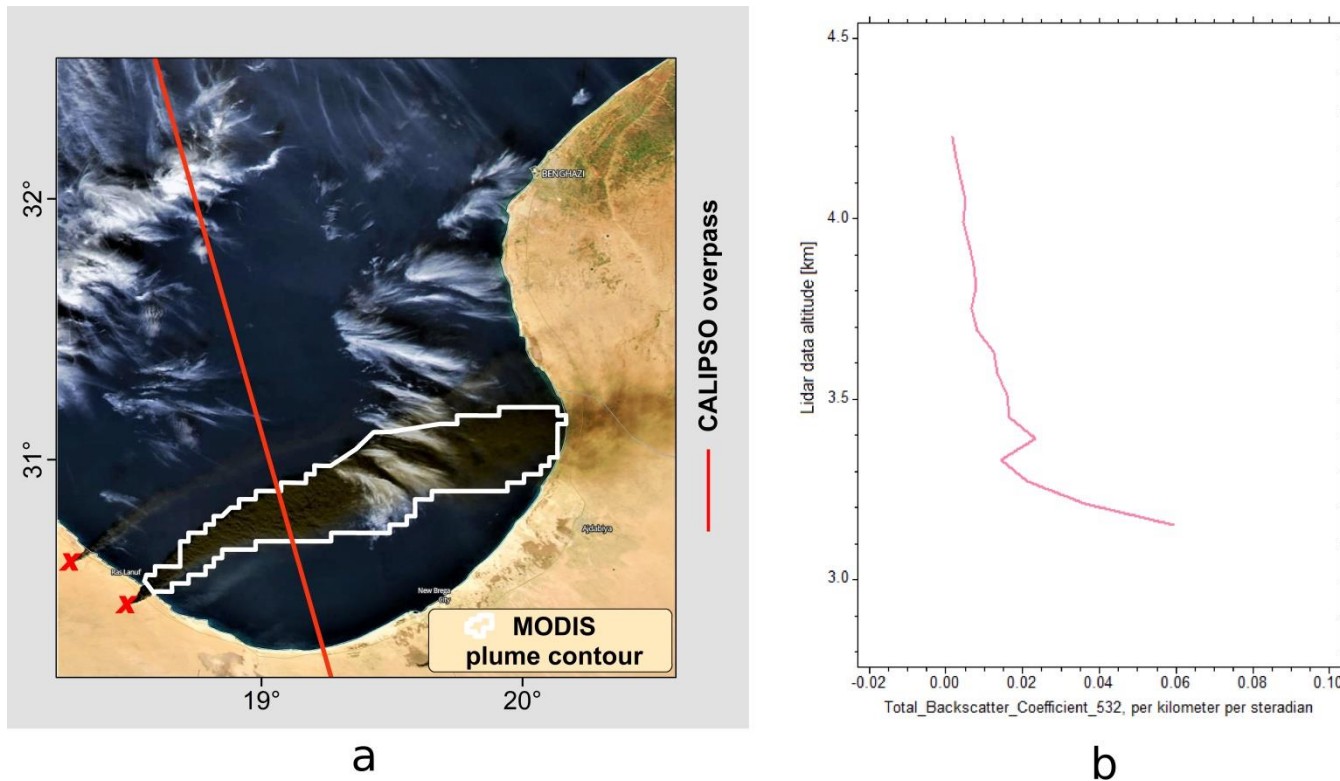

Figure 7 (a): CALIPSO overpass and MODIS plume contour; 7 (b): Single shot profile of CALIPSO level 1 Total Attenuated Backscatter
(532 nm) (Satellite imagery from the NASA Worldview application, https://worldview.earthdata.nasa.gov).




Figure 8 a) Image of Ra's Lanuf plume, 6th of January 2016, based on CALIPSO Total Attenuated Backscatter (532 nm) vs Lidar data altitude data; b) Aerosol feature classification; c) cloud feature classification






Judging from these images and from the average CAD score of – 48, the smoke plume represented a mixed feature of cloud and aerosols. This is to be expected as water vapour and particulate matter are primary components in emissions of petrochemical burnings (Johnson et al., 1991; Ferek et al., 1992; Daum et al., 1993). Cloud formations on top of oil smoke plumes have been observed in other instances (Johnson et al., 1991), as seen in figure 9, a phenomenon which hinders AOD

retrievals in both passive and active sensors.





Figure 9. Cloud formation on top of oil smoke plumes. Upper images depicting the fire at Balogan, Indonesia, 29 March, 2021; Lower left image depicting the fire at Vasylkiv, Ukraine on the 9th of June, 2015; Lower right image depicting the fire at Butcher Island, India on the 7th of October, 2017 (Satellite imagery from OpenStreetMap © OpenStreetMap contributors, 2021. Distributed under the Open Data Commons Open Database License (ODbL) v1.0 and Planet Team, https://www.planet.com/).



The current version of the Vertical Feature Mask gives a mixed result to aerosol typing comprised of dust, polluted dust and smoke aerosols for this oil smoke plume. Table 5 shows large values for AOD ranging from 1.52 (532 nm) to 1.43 (1064 nm). The low background AOD values indicate a clean atmospheric scene, thus the large AOD values are considered

to be related to the smoke plume. We also computed the AE (532/1064) which resulted in 0.09, indicating the presence of coarse particles. It should be mentioned that this event was an optimal case for constrained lidar ratio retrieval since the feature was surrounded by clear air. However, due to the fact that the plume feature was completely opaque, the lidar ratio could not be obtained from a constrained solution via the two-way layer transmittance.

Table 4. Backscatter and extinction statistics for plume values based on CALIPSO lidar measurements

| | Particulate backscatter (plume) | | | | | | (background) | |
| Event | mean 532 | STDEV 532 | STER 532 | mean 1064 | STDEV 1064 | STER 1064 | mean 532 | mean 1064 |
|---|---|---|---|---|---|---|---|---|
| 06.01.2016 | 0.015 | 0.016 | 0.002 | 0.017 | 0.016 | 0.002 | - | - |
| 01.07.2016 | 0.006 | 0.003 | 0.0004 | 0.005 | 0.002 | 0.0004 | 0.002 | 0.001 |
| 17.07.2016 | 0.007 | 0.002 | 0.0004 | 0.007 | 0.004 | 0.0008 | 0.002 | 0.002 |
| 22.08.2008 | 0.007 | 0.004 | 0.0006 | 0.008 | 0.005 | 0.0007 | 0.001 | 0.0009 |
| 29.12.2014 | 0.002 | 0.001 | 0.0005 | 0.002 | 0.001 | 0.0007 | 0.0009 | 0.001 |
| 21.10.2016 | 0.014 | 0.011 | 0.003 | 0.014 | 0.012 | 0.003 | 0.003 | 0.004 |
| | extinction coefficient (plume) | | | | | | (background) | |
| Event | mean 532 | STDEV 532 | STER 532 | mean 1064 | STDEV 1064 | STER 1064 | mean 532 | mean 1064 |
| 06.01.2016 | 1.659 | 1.823 | 0.268 | 1.554 | 1.588 | 0.234 | - | - |
| 01.07.2016 | 0.312 | 0.155 | 0.022 | 0.238 | 0.129 | 0.018 | 0.116 | 0.090 |
| 17.07.2016 | 0.314 | 0.122 | 0.021 | 0.320 | 0.212 | 0.037 | 0.131 | 0.089 |
| 22.08.2008 | 0.435 | 0.253 | 0.035 | 0.419 | 0.264 | 0.037 | 0.076 | 0.046 |
| 29.12.2014 | 0.105 | 0.043 | 0.021 | 0.099 | 0.055 | 0.027 | 0.045 | 0.035 |
| 21.10.2016 | 0.733 | 0.621 | 0.179 | 0.662 | 0.567 | 0.163 | 0.175 | 0.180 |

Table 5. Mean plume values of aerosol optical properties based on CALIPSO lidar measurements.

| Event | Plume AOD 532 | background AOD 532 | Plume AOD 1064 | background AOD 1064 | Plume AE 532/1064 | background AE 532/1064 | Length of cross section (5 km resolution) | Height of cross section 0.06 km resolution) |
|---|---|---|---|---|---|---|---|---|





| 06.01.2016 | 1.526 | Clear air | 1.430 | Clear air | 0.09 | - | 15 | 0.920 |
|---|---|---|---|---|---|---|---|---|
| 01.07.2016 | 0.046 | 0.017 | 0.035 | 0.013 | 0.39 | 0.38 | 100 | 0.150 |
| 17.07.2016 | 0.084 | 0.035 | 0.086 | 0.024 | -0.03 | 0.54 | 35 | 0.274 |
| 22.08.2008 | 0.163 | 0.028 | 0.157 | 0.017 | 0.05 | 0.71 | 40 | 0.375 |
| 29.12.2014 | 0.025 | 0.008 | 0.023 | 0.010 | 0.12 | -0.32 | 5 | 0.240 |
| 21.10.2016 | 0.088 | 0.021 | 0.079 | 0.021 | 0.15 | 0.03 | 30 | 0.120 |

Table 6. Mean plume values for lidar specific aerosol properties (PDR – particulate depolarization ratio; Lidar ratio) based on CALIPSO measurements.

| Event | PDR 532 nm | Background PDR 532 nm | Lidar ratio 532 nm | Lidar ratio 1064 nm |
|---|---|---|---|---|
| Ra's Lanuf, Libya 22.08.2008 | 0.11 | 0.17 | 55 | 48 |
| As Sidr, Libya 21.12.2014 | 0.12 | 0.18 | 37 | 37 |
| Ra's Lanuf, Libya 06.01.2016 | 0.11 | - | 109 | 86 |
| Qayyara, Iraq 01.07.2016 | 0.27 | 0.25 | 44 | 44 |
| Qayyara, Iraq 17.07.2016 | 0.32 | 0.19 | 44 | 44 |
| Qayyara, Iraq 10.21.2016 | 0.15 | 0.22 | 47 | 45 |

Within the 12-year period we identified two more cases in the Gulf of Sidra. The event at As Sidra on 29 December 2014 is an example of an unconstrained lidar ratio solution. In this case, as with all the remaining cases, CALIPSO night-time overpasses do not allow for a direct comparison with MODIS data. The plume at As Sidr can be observed in figure 10a. Off the coast of As Sidr a distinct feature above the sea surface reaching 650 m in altitude is observed. Figure 10b show a particulate backscatter profile where we can distinguish a plume thickness of approximately 240 m. This event was smaller in magnitude with respect to the Ra's Lanuf event where multiple storage tank fires contributed to the same plume mass. In this case, CALIPSO overflew much closer to the tank farm also resulting in a narrower plume cross section. The SIBYL algorithm failed to detect the plume area as a specific feature and thus, level 2 products were averaged over a larger 20 km area, as opposed to the 5 km averaging resolution, in which background aerosol levels were significantly lower. This was expected as the plume cross section measured approximately 3 km, as seen in figure 10c and d. Consequently, CALIPSO identified dusty marine aerosols within the Gulf region as evident from the lidar ratio of 37 sr (both 532 and 1064 nm) within the atmospheric layer (Kim et al., 2018). The plume particulate depolarization ratio value, 0.12 was close to local background values of 0.18 and similar to the values from Ra's Lanuf event, 0.11. Although this value is similar to the previous event, it should be mentioned that this value is indicative of the larger 20 km feature and not specifically to the





smaller 3 km plume feature which was not successfully identified by the SIBYL algorithm. The local scene over the Gulf region was fairly clean judging by the low AOD values seen in table 5. The low plume AOD value can be a result of the larger averaging resolution within the background layer. Another reason for low AOD retrievals was proposed by Jethva et al. (2014) (Jethva et al., 2014) and was shown in Deaconu et al. (2017) (Deaconu et al., 2017). In case of optically thick

aerosol layer, the sensitivity of the backscattered signal would be reduced or lost because of the strongly attenuated two-way transmission. As a result, the operational algorithm may position the base of the aerosol layer higher in altitude, thus underestimating the geometrical thickness of the aerosol layer and consequently the AOD. The selection of an inappropriate aerosol lidar ratio might also contribute to the underestimation of the AOD. AE values were also relatively low, 0.12 however these are again not indicative of plume optical properties as they are for the larger background layer. In any case,

computing AE based on low AOD values may not be a good estimate for local aerosol particle size. Figure 10d shows the same plume composition as in the case of the Ra's Lanuf plume. The vertical feature mask shows a mixt feature of clouds, aerosols and low confidence aerosol. Figure 10c, shows the aerosol classification within the aerosol layer. Judging by these results the aerosol layer reaching up to 1.5 km above mean sea level was classified as a mixture of dust, polluted dust and smoke. The plume cross section was successfully identified as smoke aerosols however the background layer situated north

of the plume was not typed as dusty marine, as seen from the lidar ratio of 37 sr.











Figure 10 Image of As Sidr, December 29<sup>th</sup> 2014 (a) based on CALIPSO Total Attenuated Backscatter (532 nm) vs Lidar data altitude data. b) Single shot profile of CALIPSO level 1 Total Attenuated Backscatter (532 nm) at As Sidr. c) Aerosol feature classification and d) cloud feature classification at As Sidr. e) Image of Ra's Lanuf plume, August 22<sup>nd</sup>, 2008 (c) plume, based on CALIPSO Total Attenuated
Backscatter (532 nm) vs Lidar data altitude data. f) Single shot profile of CALIPSO level 1 Total Attenuated Backscatter (532 nm) at Ra's Lanuf. g) Aerosol feature classification and h) cloud feature classification at Ra's Lanuf.

Another event in the region was captured inland as the plume cross section was identified 170 km south of the Ra's Lanuf tank depot. Figure 10e shows a particulate backscatter profile through the plume center describing a fairly inhomogeneous mass of smoke particles. The main plume was concentrated between 500 and 900 meters however lower
concentration may have been mixed with local dust particles all the way up to 1500 meters. Figure 10f shows the extent of the plume as it travelled southwards inland. As was the case of the previous event, plume lidar ratios were determined by an unconstrained solution. Thus values of 55 and 48 sr from 532 and 1064 nm channels were determined within the plume feature as well as for the entire layer within figure 10.e scene representing the local PBL. These values suggest a mixture of polluted dust within the local PBL (Kim et al., 2018). Plume particulate depolarization ratio values of 0.11 are similar to the
previous Ra's Lanuf event while background values of 0.17 are indicative of a polluted dust mixture (Omar et al., 2009). The average plume particulate backscatter (532 nm) measured 0.007 $km^{-1}sr^{-1}$ while the 1064 nm channel measured 0.008 $km^{-1}sr^{-1}$ which showed an increase of 6 to 9 times larger than the local background values. This large difference is also evident from the AOD values since averages of 0.16 (532 nm) and 0.15 (1064 nm) were 5 to 9 times higher than the local background levels, however these plume values were directly influenced by the lidar ratio solution. Thus, a constrained solution may
have resulted in larger values since smoke LR values are generally higher than polluted dust values (Kim et al., 2018). AE values remain low 0.05 and similar to the previous event at Ra's Lanuf suggesting a coarse dominant aerosol mixture, while background values were more indicative of polluted dust, averaging 0.71. Unlike the As Sidr event, this plume was correctly identified by the SYBIL algorithm thus resulting in plume AOD values higher than background values. Similar to the previous events the feature classification algorithm shows a mixture of clouds and aerosols in the plume bins. This
composition is evident in figure 10h and may affect the retrieval of smoke optical properties. Figure 10g shows the local aerosol layer up to approximately 2 km above local ground level. A significant portion of plume bins were discarded as cloud features even though the CAD score of – 99 indicated high confidence aerosols.







Figure 11 a) Image of Qayyara plume, July 1, 2016 based on CALIPSO Total Attenuated Backscatter (532 nm) vs Lidar data altitude data. b) Single shot profile of CALIPSO level 1 Total Attenuated Backscatter (532 nm) and c) Aerosol feature classification, July 1, 2016. d) Image of Qayyara plume July 17, 2016 based on CALIPSO Total Attenuated Backscatter (532 nm) vs Lidar data altitude data. e) Single shot profile of CALIPSO level 1 Total Attenuated Backscatter (532 nm) and f) Aerosol feature classification, July 17, 2016. g) Image of Qayyara plume, October 21, 2016 based on CALIPSO Total Attenuated Backscatter (532 nm) vs Lidar data altitude data. h) Single shot 615 profile of CALIPSO level 1 Total Attenuated Backscatter (532 nm) and i) Aerosol feature classification, October 21, 2016.

The events at the Qayyara oil fields in Norther Iraq were captured by CALIPSO in three distinct cases. In all three cases CALIPSO overpassed within less than 35 km southwest from the well fires. The first case was captured on 1 July 2016 as seen in Figure 11a. The plume can be observed just above the local surface travelling southwards and spanning over 100 km. Figure 11b shows the plume elevation which was identified from approximately 250 m (local surface elevation) to 620 around 400 m above sea level averaging 150 m in thickness. The plume particulate backscatter coefficient values ranged from 3 (532 nm) to 5 (1064 nm) times higher than local background values. Since the plume was identified within the PBL, a



lidar ratio of 44 sr for both channels was assigned indicating the presence of dust aerosols. Particulate depolarization ratio values were higher than the previous cases, on average 0.27 for plume and 0.25 for background, suggesting dust dominance. AOD values observed in plume the bins are 2.7 times higher than the local background. As with the previous cases the AE

values remain consistently low 0.39 indicating the presence of coarse dominant aerosol mixture. Figure 11c shows the aerosol classes identified in the local scene. The results indicate a dust dominant mixture with no obvious smoke signatures. As opposed to the previous cases in Norther Libya, the vertical feature mask did not indicate the presence of clouds features in the plume bins. This may be caused by low relative humidity levels and high percentage of dust particles within the mixt feature.

The second case was observed on 17 July 2016 (figure 11d) locating the plume bins closer to the burning oil wells. The plume was much narrower than the previous case however particulate backscatter coefficient values remained similar for both cases, 0.007 km$^{-1}$sr$^{-1}$ (532 and 1064 nm). Figure 11d shows the plume top height reaching 450 m above sea level with elevated particulate backscatter levels in the first 3 bins above local ground level, figure 11.e. As with the previous case the plume was contained within the PBL and a lidar ratio of 44 sr for both channels was assigned indicative of dust particles.

The particulate depolarization ratio showed large values, i.e., 0.32, which indicate a dust dominated scene. Plume AOD values were on average 2 to 3 times higher than local background values while AE remained low at -0.03 similar to the previous events. The feature classification algorithm did not identify a cloud presence in the plume bins, with figure 11.f showing the presence of dust particles.

The last event retrieved was very different to the previous two events at Qayyara. On 21 October, 2016 the black

smoke plumes from the oil fields were mixed with a thick SO$_2$ plume as a result of the Islamic State setting fire to the Al-Mishraq Sulphur plant situated NNE of the burning oil fields. Figure 11g shows the oil smoke plume over the Qayyara oil fields with average particulate backscatter coefficient (532 and 1064 nm) values of 0.14 km$^{-1}$sr$^{-1}$. Judging by figure 11h and 10i we can deduce that a mixed aerosol layer was distributed between 1500 and 2600 m. This mixed layer was also suggested by (Kahn et al., 2019) who used MISR Active Aerosol Plume-Height (AAP) to establish the SO$_2$ and oil smoke

plume elevation on the same day (Khan and Zhaoying, 2020). Regarding the plume close to the surface, which we attribute to be oil smoke, the particulate backscatter (532 nm) plume values were 3.9 times higher than local background values. The smaller oil plume was observed, on average, 120 m above the local surface levels. Lidar ratios of 47 sr and 45 sr (532 and 1064 nm) for the PBL containing the plume suggested the presence of dust particles. The plume particulate depolarization ratio however revealed lower values of 0.15 versus 0.22 for the background layer suggesting the mixing with smoke

aerosols. In this case the smoke plume increased the overall background values of AOD by a factor of 4. Since the unconstrained solution for the lidar ratio did not indicate the presence of the smoke plume, it is safe to say that a constrained, smoke-like solution would have resulted in higher AOD values. As with all cases previously described a low AE value of 0.15 indicated the presence of coarse dominant dust and smoke particles. The aerosol subtyping algorithm did not distinguish between oil smoke, SO$_2$ and dust particles revealing no obvious smoke signatures. This is evident in figure 10.i as the local

scene is seen dominated by dust and polluted dust.





Based on CALIPSO measurements, the smoke backscatter and extinction coefficient ranged from 2 to 9 times higher than background levels. In four out of six cases, particulate depolarization ratio revealed values between 0.11 and 0.15 resembling moderately depolarizing smoke while larger values in two cases were mostly due to the presence of dust particles in the local atmospheric scene. Apart from one case, all lidar ratios were assigned to a constant value as the plume

resided in the PBL. The opaque feature measured high lidar ratios of 109 sr (532 nm) and 86 sr (1064 nm) that resembles smoke lidar ratio found in literature (Giannakaki et al., 2016; Haarig et al., 2018). Although at 1064 nm the lidar ratio values for polluted continental and elevated smoke particles do not reach such high values 86 sr in this case we suspect the values are a strong indicator for the  heavy light absorbing nature of the smoke plume. Average CAD scores ranged from – 46 to – 99 which would indicate a strong confidence for the presence of aerosols. The feature classification algorithm indicated the

presence of small clouds in 3 out of 6 cases, suggesting mixed cloud-aerosol features. AE values were consistently low in all cases suggesting the presence of larger smoke particles in the plumes cross sections. AOD values ranged significantly as these are directly influenced by fuel burning rates, local background aerosol loading and a proper estimate of extinction coefficient based on correct lidar ratio solutions. The first Ra's Lanuf plume was identified above the PBL between 2400 and 4200 m. This is a good indicator for the magnitude of the event as it involved several tank fires with higher burning rates

simultaneously injecting larger concentrations of aerosol at higher elevations in the troposphere. Based on this small number of events it is difficult to assign a separate aerosol type for these oil smoke plumes. However valuable information regarding size distributions, particulate depolarization ratio and to some extent lidar ratio can be retained from this study. It should be mentioned that these values reflect smoke plumes located very close to the fire sources and thus present low mixing ratios with other local aerosols.

**3.3 AERONET case study**

As discussed in the introduction section, oil smoke plumes have been rarely observed using ground based remote sensing instruments such as AERONET sun photometers. We used AERONET version 3 direct sun data to assess the presence of oil smoke plumes. Only one study was found in scientific literature (Mather et al., 2007) which measured aerosol properties of the Bouncefield plume at two distinct locations. Here we identified the smoke plume resulting from naphtha

tank fires in Vasylkiv, Kyiv Oblast, Ukraine on 9 June 2015. The smoke plume was also captured in RGB images as seen in figure 6g. Figure 12a shows the distinct signature of the oil smoke plume as AOD values increased significantly in all wavelengths. Figure 12c is a good indication of the increasing particle size with respect to the other days observed as well in MODIS and CALIPSO data. Figure 12d shows the daily evolution of AE with values between 0.45 and 0.9 for the time frame in which the plume was observed.  Figure 12b shows AOD values rising as the plume was travelling NE over Kiev.

The AERONET station in Kiev is situated approximately 35 km NE of the Vasylkiv tank farm. The peak of the plume was detected at 9:45 UTC when the AOD was 0.68 at 500 nm. Unfortunately, no inversion products coinciding with direct sun measurements were available as the Kiev sky was partially cloudy at the time.





Figure 12. AOD and AE smoke plume values at Kiev on 9 June , 2015 and monthly values from June 2015

## 3.4 Data comparison between methods and other similar studies

Table 7. Oil smoke optical properties from ground based and flight measurements along with the scientific reference.

| Aerosol properties | Reference | Comments |
|---|---|---|
| | *Gulf War smoke plumes* | |
| AOD: 0.82 – 1.92 | (Pilewskie and Valero, 1992) | Visible optical depths |
| AOD: 1.5 ; AE: 0.7 | (Nakajima et al., 1996) | 0.5 μm |
| AOD: 2.0 – 3.0 | (Hobbs and Radke, 1992) | Visible optical depths |
| PDR: 0.14 | (Okada et al., 1992) | 532 nm (lidar) |





| AOD: 0.2 -0.6 Lidar ratio: 38 sr | (Ross et al., 1996) | 532 nm (lidar) |
|---|---|---|
| Extinction coefficients: 0.06 - 1.30/km; 0.06 -1.60/km AOD: 0.05 – 1; 0.05 – 1.20 | (Laursen et al., 1992) | 1064 and 532 nm (lidar) |
| Extinction coefficients: 0.5 – 10/km | (Weiss and Hobbs, 1992) | 538 nm (photometer) |
| AOD: 0.5 | (Johnson et al., 1991) *Bouncefield smoke plume* | 11 µm |
| Near source: AOD 0.4 – 1.4; AE 0.42; $R_{eff}$ 0.45 – 0.85 µm Distant location: AOD 0.3 – 0.7 AE 0.09; $R_{eff}$ 0.88 – 1.40 µm | (Mather et al., 2007) | 440, 675, 870, 936 and 1020 nm (sun photometer) |

The results presented in this study show a wide range of values which are attributed to a multitude of local factors such as: background aerosols, burning rates, weather conditions, fuel type, time of retrieval, and local geography. Other factors can be attributed to the different types of methods and algorithms used to retrieve aerosol specific data. MODIS data showed relatively low values of plume specific AOD ranging from -0.04 to 0.16. The only event which was captured by both MODIS and CALIPSO retrievals showed a large level of discrepancy. In particular, the Ra's Lanuf event on 6 January 2016

showed average column AOD values of 0.21 over the plume area with a maximum pixel value of 0.32 (550 nm). In contrast, CALIOP measurements revealed average plume AOD values of 1.52 (532 nm) which where plume specific as no other extinction values were detected beneath or above the plume through the troposphere and stratosphere in the local scene. In the remaining 5 cases, CALIPSO retrieved AOD values ranging from 0.02 to 0.16 for average plume thickness ranging from 0.120 to 0.375 km. While these values more closely resemble the successful MODIS retrievals one should restrain from a

forward comparison.  A reason is that neither of these events were analysed by both sensors since MODIS did not successfully retrieved AOD values over land. This coupled with high levels of uncertainty surrounding MODIS LUT values and CALIPSO lidar solutions suggest the need for a more in depth analysis. The one case seen by an AERONET sun photometer indicates AOD values ranging from 0.3 to 0.68 (500 nm) however the satellite images suggest that these values were not indicative for the main plume which most likely did not reached Kiev. Nevertheless, MODIS did not successfully

retrieve any AOD values from this event or any other over land while for other events over ocean it did not yield such high AOD values. It is safe to say that MODIS AOD retrievals for oil smoke plumes may not produce satisfactory results since the predetermined LUT values may not contain similar events to the ones described in this study. The CALIPSO AOD measurements are directly influenced by the lidar ratio. For these specific events, a correct estimate of lidar ratio is very difficult to achieve based on unconstrained solution. On one hand these lidar ratios are not directly measured.  On the other

hand, lidar ratio for oil smoke plumes may exhibit a different behaviour, considering the high BC content, different from biomass smoke or smoke/polluted continental aerosols. In cases of "clean" atmospheric conditions a constrained solution may result in better AOD estimates however these conditions are rarely achieved, with less than 0.01% of all aerosol layers detected (Tackett et al., 2018). AE values seem to be more consistent between MODIS, CALIPSO and AERONET as all





techniques suggest the presence of coarse aerosol mixtures, however in conditions with low AOD values, one should restrain
from direct comparisons.

Table 7 lists the oil smoke optical properties from different studies that on average indicate much larger AOD values. It should be mentioned that AOD values from the Gulf war smoke plumes are larger for the most part due to the magnitude of the event. These measurements describe super composite plumes resulted from o large number of well fires and pool fires. An event that more closely resembles the events in this study was analysed by Mather et al. (2007) who
retrieved AOD values of 0.3 to 0.7 50 km away from the oil depot (Mather et al., 2007). These values are similar to the AERONET values from Kiev presented in this study as in both cases AOD was retrieved using sun photometers and the Kiev AERONET station is located at approximately 20 km from the oil depot. However, the Bouncefield event was significantly larger than the Vasylkiv event. CALIPSO AOD measurements from Ra's Lanuf (January 6, 2016), are similar to other studies described in table 7 while AE values are in general in good agreement with our case studies. Judging by the
results seen in figure 8.a, an aerosol feature exhibiting such heavy attenuation in the layers directly beneath would most likely yield higher AOD values. $R_{eff}$ values from MODIS also show the presence of larger particles analogous to Mather et al. (2007). Particulate depolarization ratio for four out of six cases reflects the values shown by Okada et al. 1992 indicating that oil smoke particles are moderately depolarizing. The opaque feature indicated high lidar ratio values (109 sr) much larger than (Ross et al., 1996). This may be a result of plume dimensions as the Ra's Lanuf plume cross section was much
larger and thicker than the plume described in Ross, 1996. One would expect large lidar ratio values to be the result of the highly absorbent nature of these smoke plumes (high percentage of black carbon, high plume homogeneity and low mixing ratio) leading to larger extinction values. CALIOP extinction coefficient averaged for each individual plume in this study ranged from 0.10 to 1.65 km$^{-1}$ (532 nm) and 0.10–1.55 km$^{-1}$ (1064 nm) with a maximum bin value of 9.6 km$^{-1}$. These values agree well with (Weiss and Hobbs, 1992) and (Laursen et al., 1992).

**4 Conclusions**

In this study, we examined oil smoke plumes derived from 30 major industrial events within a 12 year period. To our knowledge this is the first study that utilized a synergetic approach based on satellite remote sensing techniques. The MODIS ocean algorithm was successful in retrieving aerosol properties which, on average, ranged from -0.06 to 0.16 for plume specific AOD, -0.18 to 1.25 for Angstrom exponent and 0.29 to 1.73 µm for effective radius. Apart from the SOCAR
event, all the remaining smoke plumes exhibited AE values lower than 0.74 suggesting that the smoke plumes were coarse mode dominant. CALIPSO measurements showed values of plume AOD ranging from 0.02 to 0.16 (532 nm) and 0.02 to 0.15 (1064 nm) except for one event where AOD values reached 1.52 (532 nm) and 1.43 (1064 nm). AE values ranged from -0.03 to 0.39 which agree with MODIS. A large discrepancy can be found in one event (at Ra's Lanuf, Lybia, on the 6[th] of January 2016) where CALIPSO measured column integrated AOD values 5 times higher than MODIS. For this event
CALIPSO produced a retrieval that resulted in high lidar ratios of 109 sr (532 nm) and 86 (1064 nm). The high concentration



of water vapour emitted by the oil fire may have contributed to instances of small cloud formation above the smoke plume and thus contaminating the retrievals. Typically, lidar ratio ranged from 37 to 55 sr (532 nm) and 37 to 48 sr (1064 nm) however these unconstrained solutions were indicative of the local aerosol scene and not directly measured. Particulate backscatter coefficient values ranged from 0.002 to 0.015 $km^{-1}sr^{-1}$ (532 nm) and 0.002 to 0.017 $km^{-1}sr^{-1}$ (1064 nm).

Particulate extinction coefficient values ranged from 0.10 to 1.65 $km^{-1}$ (532 nm) and 0.10 to 0.155 $km^{-1}$ (1064 nm). On average backscatter and extinction coefficient values were 2 to 9 times higher than the local background. Particulate depolarization ratios ranged from 0.11 to 0.15 for four out of six cases while the remaining cases were 0.27 and 0.32. We suspect that this discrepancy in the two cases at Qayyara are a result of dust aerosols presented in the smoke plumes. The values presented agree with similar studies that used ground-based and airborne measurements. We believe that MODIS

gives a conservative estimate of the plume AOD since MODIS algorithms rely on general aerosol models and various atmospheric conditions within the look-up tables which do not reflect the highly light absorbent nature of these smoke plumes. Furthermore, the spectral reflectance relationship used by MODIS algorithms may hinder most retrieval attempts as thick black plumes exhibit a distinct spectral signature. CALIPSO measurements are heavily dependent on unconstrained solutions, which in turn do not reflect the oil smoke plumes. Thus, we also believe that the AOD values based on CALPSO

measurements are conservative in nature since strong absorbing smoke would yield larger lidar ratios and AOD values. We stress the need for further lidar measurements of oil smoke plumes since, based on this study we cannot conclude whether these aerosols belong to a different smoke subtype. Future space-borne lidar mission such as EarthCare (Illingworth et al., 2015) will provide direct measurements of lidar ratios and the possibility of better AOD estimations with regard to these types of events. Based on this study we concluded that the MODIS land algorithms are not yet suited for retrieving aerosol

properties for these types of smoke plumes due to the highly light absorbing properties of these aerosols. This study has shown a novel method of oil smoke plume identification and analysis which does not require, in some cases, perilous field work. We believe that these types of studies are a strong indication for the need of improved aerosol models and retrieval algorithms. For these types of aerosols, better AOD estimates are important for both air quality and climate change implications.

**Code availability**

Not applicable.

**Data availability**

CALIPSO data is available at:

Winker, David: CALIPSO LID L1 Standard HDF File - Version 4.10, https://doi.org/10.5067/CALIOP/CALIPSO/LID_L1-

STANDARD-V4-10, 2016.



Winker, David: CALIPSO Lidar Level 2 5 km Aerosol Layer Data V4-20, https://doi.org/10.5067/CALIOP/CALIPSO/LID_L2_05KMALAY-STANDARD-V4-20, 2018a.

Winker, David: CALIPSO Lidar Level 2 Aerosol Profile Data V4-20, https://doi.org/10.5067/CALIOP/CALIPSO/LID_L2_05KMAPRO-STANDARD-V4-20, 2018b.

Winker, David: CALIPSO Lidar Level 2 Vertical Feature Mask Data V4-20, https://doi.org/10.5067/CALIOP/CALIPSO/LID_L2_VFM-STANDARD-V4-20, 2018c.

MODIS data is available:

MODIS Atmosphere Science Team: MODIS/Terra Aerosol 5-Min L2 Swath 10km, https://doi.org/10.5067/MODIS/MOD04_L2.061, 2017a.

MODIS Atmosphere Science Team: MYD04_L2 MODIS/Aqua Aerosol 5-Min L2 Swath 10km, https://doi.org/10.5067/MODIS/MYD04_L2.061, 2017b.

**Author contribution**

AM, NA and AO carried out the conceptualisation and methodology. AM, AR and HS carried out the formal analysis. AM, AR and HS provided visuals for the paper. AM, NA and CB wrote the initial draft. NP, LD and DN reviewed

and edited the initial draft. NA and CB acquired funding for the current research. AO provided supervision for the PhD students AM and AR.

**Acknowledgements**

This work was supported by the Project entitled "*Development of ACTRIS-UBB infrastructure with the aim of contributing to pan-European research on atmospheric composition and climate change*" SMIS CODE 126436, co-financed

by the European Union through the Competitiveness Operational Programme 2014 – 2020.

This work was supported by the Project entitled "*Strengthening the participation of the ACTRIS-RO consortium in the pan-European research infrastructure ACTRIS*" SMIS CODE 107596, co-financed by the European Union through the Competitiveness Operational Programme 2014 – 2020.

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
