# Peer review of "A novel method of identifying and analysing oil smoke plumes based on MODIS and CALIPSO satellite data"

_Atmospheric Chemistry and Physics, 2021_

## Referee Comment (RC2)

**Review of paper acp-2021-970**

The paper proposes a method based on the use of MODIS and CALIPSO data to assess several parameters of oil-burning plumes. I think that the topic is relevant, but perhaps the title of the paper is misleading as one gets the feeling that, more than demonstrating the usefulness of the method, the authors reveal limitations that should be overcome before it can be customarily applied. In my opinion the paper should be revised in order to present the results in a more organized way and make clearer for which events a true synergic (MODIS and CALIPSO) study could be carried out. Just as an example: the authors list in table 1 thirty oil-burning events whose plumes have been observed by MODIS instruments, which, they claim, they have examined (line 741). It's understandable that not all the events are specifically presented, but, if they have been examined, are there general conclusions that can be drawn from these observations? How many of these events could also be observed by CALIPSO? Along the text, based on examples, some hint is given that not all the plumes exhibit similar characteristics, but, after applying the methods described, can they be classified according to some typing criterion?

**General remarks**

1. In my opinion the authors should improve the organization of the paper and help the reader in understanding the rationale of some choices made. For example, what's the reason for the order in which the events are listed in table 1 (certainly not chronological)? I would suggest that the events discussed in detail in the text are referred to by their numbering in the table; otherwise it is sometimes difficult to understand which event the authors are discussing, especially because sometimes they allude to them with a name different from the location given in table 1. For example, in line 133 it is said that "only the event in Kiev was analysed by different techniques". One has to guess that the referred event is number 14 in the actual numbering of table 1. The Caspian sea event (No. 8 in table 1) is also referred to as the Gunashli event (line 435); the reader needs to be a geography expert to realize that the authors are talking about the same event.

2. Which events have been studied in a really synergic (MODIS + CALIPSO) way? Only one? ("The only event which was captured by both MODIS and CALIPSO retrievals showed a large level of discrepancy", lines 698-699). If this is so, I would strongly suggest that the title is modified. Both the MODIS-based and the CALIPSO-based analyses are worth being presented, but in that case the "synergic" term in the title would be too strong in my view.

3. The rationale after which some events are discussed in particular should be given to help the reader understand the reason why these events, and not others, are discussed. For example, the case of Ra's Lanuf and As Sidr is properly introduced ("We selected a successful retrieval to better describe the method used for our analysis", line 342), as it is the SOCAR Platform in the Caspian Sea ("Next, we examined the smoke plume from SOCAR's Platform No.10 fire in the Caspian Sea as an "atypical" event based on the fuel type and plume albedo", lines 374-375), but the reason why

the next events described in section 3.1 (Deepwater Horizon, JX Nippon, Ra's Lanuf again, Puerto Sandino, Escravos…) are chosen is unclear.

**Main specific remarks**

1. I think that the abstract is too specific in giving values of AOD and AE obtained from MODIS and CALIPSO and discussing the discrepancies between MODIS and CALIPSO retrievals for the (yet unidentified in the abstract) event of Ra's Lanuf on 6[th] of January 2016. Although this is debatable, in my opinion the abstract should highlight the contributions of the method, rather than the results obtained.

2. In section 1.2 (Event synopsis), why only a few events are discussed? What's the rationale for their selection?

3. The sentence in lines 302-304 sounds confusing: "Focusing on the plume bins we used the extinction coefficient to backscatter ratio to determine the plume averaged lidar ratios if the plume was not identified as a distinct aerosol feature." But the extinction coefficient to backscatter [coefficient] ratio is the lidar ratio. The explanation looks circular.

4. The events in table 2 and 3 are difficult to identify based only on the date. Note in addition that in table 3 there is a row more than in table 2. Didn't MODIS in Terra retrieve values for the event of 04.01.2018? The same remark on the inconvenience to identify events based on the date apply to tables 4 and 5. Moreover, in these tables the events coincident with events in table 2 and 3 should be pointed out. The event on 21.10.16 doesn't seem to be listed in table 1. Is the date correct? Should it be 21.01.16?

5. Apart from the issues mentioned in point 3, I had a lot of trouble following section 3.1. The initial discussion (lines 342 – 362) refers to figure 2. However, in lines 362 – 363, tables 2 and 3 are referred without saying that they include data from other locations, described in figure 3 and in the subsequent text. In addition, figure 3c is quoted twice ("Figure 3c shows a plume specific AOD values 390 ranging from 0.06to 0.23", lines 390-391; "Figure 3d shows AOD values as high as 0.24 over the average AOD background level for the plume originating at Ra's Lanuf.", lines 391-392), but the second time should be probably figure 3d. Then figure 3d is cited again in line 394. It would be better to group together all the discussion referred to given event. One has to guess, moreover, that the Ra's Lanuf oil field pertains (probably, not sure) to the Surt district of Lybia (line 394).

6. In figure 3, the color scale seems to correspond to the plume specific AOD. This should be indicated in the legend of the color scale to avoid confusion with the total AOD. Please check also what is represented in the color scales of figure 6.

7. Perhaps related to the last part of point 5 above, an event in As Sidr is mentioned in table 1 (No. 19), but with no date. Is it the same event as No. 20?

8. Referring to figure 6, it is said (line 476) that "The cases form Saudi Arabia, Iran and Iraq show no values retrieved over the plume areas". What's the reason to say that in the case of panels a and b? Where should the reader expect the plume? By the way it would be helpful to say that these cases correspond to figures 6a, 6b and 6c.

9. Relating to the Ra's Lanuf plume on 6[th] of January, there are statements on the retrieved lidar ratio values that should be clarified: it is said on the one hand that the lidar ratios of 109 sr at 532 nm and 86 sr at 1064 nm (lines 513-514 and table 6) are obtained. Then, in lines 541-543 it is stated that: "It should be mentioned that this event was an optimal case for constrained lidar ratio retrieval since the feature was surrounded by clear air. However, due to the fact that the plume feature was completely opaque, the lidar ratio could not be obtained from a constrained solution via the two-way layer transmittance". This should be clarified as, at first sight, it looks contradictory. I suggest that the CALIPSO retrievals of lidar ratio on opaque layers is briefly discussed or, at least, mentioned and referenced.

**Minor specific remarks**

1. Line 30: "agreeance". I couldn't find that word in the dictionary. Probably the authors mean "agreement".

2. Lines 41-42: "CALIPSO measurements are heavily dependent on lidar ratios which are not directly measured if plumes within the planetary boundary layer". The sentence sounds strange. Perhaps "are" is missing after "if"? Please check.

3. Lines 81-82: "Most sensors cannot retrieve a wide variety of aerosol properties thus relying on inversions techniques and complex radiative transfer computations". The meaning of this sentence is unclear. Can or cannot the sensors retrieve aerosol properties? Please check.

4. In table 1, the cause of event id left blank for some events. It would be better to state "unknown" (if this is the reason for the blank space).

5. Line 188: "construct" → "construction".

6. Lines 313-314: "Based on the particulate total backscatter coefficient (532 nm) we defined the plume cross section as in each range bin, the plume values were at least 2 times higher than background values". This sentence sounds strange. Please check the wording.

7. Line 375: what does "n.d" mean in the reference "(Business-humanrights, n.d.)"?

8. Line 385: "This is evident in the plume albedo from MODIS true colour images". A figure should either be provided or said that it is not shown.

9. Lines 421-422: "Plume values close to 0 were retrieved near the event while average values registered two to three times lower than the local background". The sentence is not clear. Where are those values reported in the paper? Does this refer to the 0.34 and 0.74 values cited in line 419?

10. Lines 506-507: apparently referring to figure 7b it is said: "The average plume thickness was approximately 920 m ranging from 2700 m to 3300 m." But 3300 – 2700 = 600. In addition, the backscatter coefficient profile shown in fig. 7b does not go below 3100 m or so.

11. Lines 510-511: "average particulate backscatter (532 nm) values measured 510 0.015 km$^{-1}$sr$^{-1}$ while values at 1064 nm measured 0.17 km$^{-}$1sr$^{-}$1." The value for

1064 nm should be 0.017 km$^{-1}$sr$^{-1}$. Moreover, I think that the values measured for 532 nm and for 1064 nm are essentially indistinguishable, as they fall within the uncertainty interval of each other.

12. Table 6: are there not uncertainty intervals for the PDRs and the lidar ratios? Note in addition that the date for the Qayyara, Iraq event (10.21.2016) is certainly wrong. Why these Qayyara events not listed in table 1? Were they not observed my MODIS?

13. Line 666: "AOD values ranged significantly…" What does this mean?

14. Lines 680-681: "The smoke plume was also captured in RGB images as seen in figure 6g". But figure 6g does not seem to correspond to an RGB image.

15. Line 683: "Figure 12d shows the daily evolution of AE with values between 0.45 and 0.9". At which wavelength?

16. Line 709: "did not reached Kiev" → "did not reach Kiev".

---

## Author Comment (AC1)

**Referee comment 1.**

The paper needs an extensive language revision. It is necessary to check the writing, grammar and typos.

**Authors reply 1.**

We undertook an extensive English language revision by a native speaker.

**Referee comment 2.**

It is necessary to reduce the length of the paper, especially the results section. There are too many study cases analyzed in detail, but the main conclusions of the different analysis get lost in the text and are not clear. The number of figures should also be reduced. According to this comment, it is also necessary to improve the last part of the abstract. The authors provide a large list of numeric values, but it is not clear the message and conclusions that we can infer from these data.

**Author's reply 2.**

The overall paper has been extensively modified resulting in a general reduction of length (approx. 9 pages) and a reduction in the number of figures presented (from 12 to 9) while maintaining the overall essence of the results and findings. On the topic of detailed cases studies we focused only on the events in Libya as they were seen by both MODIS and CALIPSO. This effectively translates to 3 events detail as MODIS successful retrievals and 3 events detailed as CALIPSO retrievals. The remaining cases studies are discussed on a general basis regarding the results and conclusions of the study.

To better identify the events described in the paper and to better respond to the issues raised by the referee we updated table 1 as follows:

- We added the column "*event ID no.*", as suggested by referee 2, and in text we addressed each event by this number for consistency reasons. This column was also added to tables 2 to 7.

- The former columns entitled "Date" and "Number of observations (days)" have been merged into a single column "*MODIS observation interval*". This new column contains the dates for the "*first day*" and the "*last day*" in which MODIS RGB images showed oil smoke plumes. Some events, i.e. event 3 and 14, are listed as having more than one location. We have grouped all locations within a single event based on the fact that they share the same cause (same armed conflict) which generated the events on the same "*first day*". For these events, the coordinates for each location (oil installation involved in a fire) in particular are given.

Please find the revised version of table 1 below:

*Tab. 1 Major industrial events leading to observable smoke plumes seen in MODIS RGB images*

| Event ID No. | Location | MODIS observation interval | | Coordinates | Cause of event | Type of installation | References |
| | | Start | End | | | | |
|---|---|---|---|---|---|---|---|
| **1** | Qayyara, Iraq | 13.06.2016 | 27.03.2017 | 35.83 N ; 43.21 E | armed conflict | oil wells | (Tichý and Eichler, 2018) |
| **2** | Omidieh, Iran | 06.05.2019 | 06.05.2019 | 30.84 N ; 49.65 E | human error | oil pipeline | (Financial Tribune, 2019) |
| **3** | Haradh, Hawiyah, Uthmaniyah, Shedgum, Buqayq; Saudi Arabia | 14.09.2019 | 26.09.2019 | 24.05 N ; 49.20 E
24.80 N ; 49.35 E
25.18 N ; 49.31 E
25.64 N ; 49.39 E
25.92 N ; 49.68 E | armed conflict | oil processing | (Khan and Zhaoying, 2020)
(Reuters, 2019)
(New York Times, 2019) |
| **4** | Caspian Sea, Azerbaijan | 06.12.2015 | 18.12.2015 | 40.20 N ; 51.06 E | extreme weather | oil and gas platform | (Necci et al., 2019) |

| | | | | | | | |
|---|---|---|---|---|---|---|---|
| 5 | Gulf of Mexico, USA | 21.04.2010 | 21.04.2010 | 28.44 N ; 88.21 W | equipment failure | drilling rig | (Gullett et al., 2016) |
| 6 | East China Sea, China | 14.01.2018 | 14.01.2018 | 28.37 N ; 126.08 E | human error | oil tanker | (Li et al., 2019) (Qiao et al., 2019) |
| 7 | Houston Texas, USA | 18.03.2019 | 19.03.2019 | 29.43 N ; 95.05 E | equipment failure | storage tanks | (An Han et al., 2020) |
| 8 | Jaipur, India | 30.10.2009 | 08.11.2009 | 26.77 N ; 75.83 E | Human error | storage tanks | (Vasanth et al., 2014) |
| 9 | Sendai, Japan | 12.03.2011 | 13.03.2011 | 38.27 N ; 141.03 E | earthquake, tsunami | storage tanks | (Krausmann and Cruz, 2013) |
| 10 | Vasylkiv, Ukraine | 09.06.2015 | 10.06.2015 | 50.16 N ; 30.32 E | sabotage | storage tanks | (Kovalets et al., 2017) (Reuters, 2015) |
| 11 | Ra's Lanuf, Libya | 19.08.2008 | 25.08.2008 | 30.45 N ; 18.49 E | human error | storage tanks | (The Telegraph, 2011) |
| 12 | Ra's Lanuf, Libya | 12.03.2011 | 14.03.2011 | 30.45 N ; 18.49 E | armed conflict | storage tanks | (BBC, 2011) |
| 13 | As Sidr, Libya | 26.12.2014 | 31.12.2014 | 30.60 N ; 18.28 E | armed conflict | storage tanks | (BBC, 2014) |
| 14 | Ra's Lanuf, As Sidr; Libya | 05.01.2016 | 07.01.2016 | 30.45 N ; 18.49 E 30.60 N ; 18.28 E | armed conflict | storage tanks | |
| 15 | Surt disrtric, Libya | 14.01.2016 | 14.01.2016 | 30.02 N ; 18.50 E | armed conflict | oil pipeline | (Tichý and Eichler, 2018) (Tichý, 2019) |
| 16 | Ra's Lanuf, Libya | 21.01.2016 | 23.01.2016 | 30.45 N ; 18.49 E | armed conflict | storage tanks | |
| 17 | Ajdaviya district, Libya | 01.02.2016 | 01.02.2016 | 29.68 N ; 20.54 E | armed conflict | oil pipeline | |
| 18 | Ra's Lanuf, Libya | 17.06.2018 | 21.06.2018 | 30.45 N ; 18.49 E | armed conflict | storage tanks | (Reuters, 2018) |
| 19 | Puebla, Mexico | 19.12.2010 | 19.12.2010 | 18.96 N ; 98.45 W | illegal tapings | oil pipeline | (Biezma et al., 2020) |
| 20 | Escravos, Nigeria | 04.01.2018 | 05.01.2018 | 5.45 N ; 5.35 E | bush fire | oil pipeline | (Bloomberg, 2018) |
| 21 | Puerto Sandino, Nicaragua | 18.08.2016 | 19.08.2016 | 12.18 N ; 86.75 W | unknown | storage tanks | (Ahmadi et al., 2020) |
| 22 | Gulf of Oman | 13.06.2019 | 13.06.2019 | 25.39 N ; 57.38 E | armed conflict | oil tanker | (BBC, 2019) |
| 23 | Catano, Puerto Rico | 23.10.2009 | 24.10.2009 | 18.41 N ; 66.13 W | human error | storage tanks | (Vasanth et al., 2014) |
| 24 | Punto Fijo,Venezuela | 27.08.2012 | 27.08.2012 | 11.74 N ; 70.18 W | equipment failure | storage tanks | (Schmidt et al., 2016) |
| 25 | Butcher Island, India | 07.10.2017 | 08.10.2017 | 18.95 N ; 72.90 E | lightning strike | storage tank | (The Indian Express, 2017) |

To further address the improvements in the organization of the paper we rearranged the results section, as suggested by referee 2, to better illustrate the analysis method and the overall results. The current format of the results section includes:

Section 3.1 **Case study illustration** – we examine one event to illustrate the analysis method. This section includes the results from: one MODIS successful retrieval (event 14, ocean retrieval), one CALIPSO retrieval (event 14) and, one MODIS unsuccessful retrieval (event 13, land retrieval).

Section 3.2 **MODIS successful retrievals**

In lines 298 – 303 we address how many successful retrievals were analysed:

"*Based on the information given in table 1 we filtered a total of 375 days in which oil smoke plumes were observed by the MODIS sensors. After applying the selection criteria for the MODIS sensor we obtained a total of 10 days with successful retrievals. The majority of oil plumes resulted in unsuccessful retrievals, 70.7%, while 26.7% of plumes were screened out due to high percentage of cloud coverage. When applying the selection criteria for CALIPSO we obtained a number of 6 plume sections suitable for analysis. Table 2 shows the dates for both MODIS and CALIPSO retrievals suitable for analysis.*"

A list of MODIS successful retrievals is given in table 2 as follows:

*Table 2. List of successful MODIS retrievals and CALIPSO overpass dates*

| Event Id. Nr. | MODIS (Terra and Aqua) Successful retrieval date | CALIPSO retrieval date |
|---|---|---|

| | | 01.07.2016 |
|---|---|---|
| 1 | - | 17.07.2016 |
| | | 21.10.2016 |
| 4 | 08.12.2015 | - |
| 5 | 21.04.2010 | - |
| 9 | 11.03.2011 | - |
| 11 | - | 22.08.2008 |
| | 28.12.2014 | - |
| 13 | 29.12.2014 | 29.12.2014 |
| | 30.12.2014 | - |
| 14 | 06.01.2016 | 06.01.2016 |
| 16 | 21.01.2016 | - |
| 20 | 19.08.2016 | - |
| 21 | 04.01.2018 (only Aqua) | - |

Section 3.2 shows general MODIS results together with a more detailed discussion of the events in Lybia, event 13 and 16 (event 14 was previously discussed in section 3.1). The reasoning behind the more detailed discussion of these events is given in line 423: "*We choose to describe in detail the events from Libya as they are also analysed based on CALIPSO retrievals*"

Section 3.3 shows general results from **CALIPSO retrievals** together with detailed discussion of the events in Lybia.

Section 3.4 is now the **AERONET case study** and,

Section 3.5 is now **Data comparison between methods and other similar studies.** This section has been extensively revised to better illustrate how the methods compare to one another and to similar studies. Uncertainty intervals have been added to our results and, to the extent of which they were addressed in similar studies, uncertainty intervals were also added to the reference values in table 8:

*Table 8. Oil smoke optical properties from ground based and flight measurements along with the scientific reference.*

| | | | LIDAR | | | |
|---|---|---|---|---|---|---|
| Reference | AOD 532 nm | AOD 1064 nm | AE 550/1064 nm | PDR 532 nm | LR 532 nm (sr) | LR 1064 nm (sr) |
| This study CALIPSO | $0.025 \pm 0.010 - 1.526 \pm 0.804$ | $0.023 \pm 0.017 - 1.430 \pm 0.473$ | $- 0.03 - 0.39$ | $0.11 \pm 0.43 - 0.32 \pm 0.48$ | $37 \pm 15 - 109 \pm 47$ | $37 \pm 15 - 86 \pm 10$ |
| (Okada et al., 1992) Ground based lidar | - | - | - | $0.14 - 0.18$ | - | - |
| (Ross et al., 1996) Airborne lidar | 0.2 - 0.6 | - | - | - | 38 | - |
| (Laursen et al., 1992) Airborne lidar | $0.05 - 1 \pm 65\%$ | $0.05 - 1.2 \pm 85\%$ | - | - | - | - |
| (Ceolato et al., 2020) Ground based lidar | - | - | - | 0.058 | - | - |
| (Ceolato et al., 2021) Ground based lidar | - | - | - | - | $125.3 \pm 5.0$ sr | - |

| | Radiometer | | | Sun photometer | | |
|---|---|---|---|---|---|---|
| Reference | AOD 550 nm | AE 550/860 nm | $R_{eff}$ (µm) | AOD 500 nm | AE 440/870 nm | $R_{eff}$ (µm) |
| This study MODIS and AERONET | $- 0.04 - 0.16 \pm (0.05 + 0.20 \times AOD)$ | $- 0.18 - 1.25$ | $0.29 - 1.73$ µm | $0.28 - 0.68 \pm 0.01$ | $0.45 - 0.90$ | - |
| (Pilewskie and Valero, 1992) Airborne radiometer | $0.82 - 1.92 \pm 2\%$ (500 nm) | - | - | - | - | - |

| | | | | | |
|---|---|---|---|---|---|
| (Nakajima et al., 1996) | - | - | - | 1.5 | $0.7 \pm 2.5\%$ | - |
| (Mather et al., 2007) | | | | 0.3 – 1.6 (440 nm) | 0.09 – 0.42 | 0.45 – 1.40 μm |

To address the referee's concern regarding the improvement of the abstract we revised the section in question to better reflect the contributions of the method and the overall findings of the study. Lines 23 – 37 reflect these changes:

*The analysis method in this study was developed to better differentiate between oil smoke aerosols and the local atmospheric scene. We present several aerosol properties in the form of plume specific averaged values. We believe that MODIS values are a conservative estimation of plume AOD since MODIS algorithms rely on general aerosol models and various atmospheric conditions within the look-up tables which do not reflect the highly absorbing nature of these smoke plumes. Based on this study we conclude that the MODIS land algorithms are not yet suited for retrieving aerosol properties for these types of smoke plumes due to the strong absorbing properties of these aerosols. CALIPSO retrievals rely heavily on the type of lidar solutions showing discrepancy between constrained and unconstrained retrievals. Smoke plumes identified within a larger aerosol layer were treated as unconstrained retrievals and resulted in conservative AOD estimates. Conversely, smoke plumes surrounded by clear air were identified as opaque aerosol layers and resulted in higher lidar ratios and AOD values. Measured lidar ratios and particulate depolarization ratios showed values similar to the upper ranges of biomass burning smoke. Results compare well with studies that utilized ground-based retrievals, in particular for Ångström exponent (AE) and effective radius ($R_{eff}$) values. MODIS and CALIPSO retrieval algorithms disagree on AOD ranges, for the most part, due to the extreme light absorbing nature of these types of aerosols. We believe that these types of studies are a strong indicator for the need of improved aerosol models and retrieval algorithms.*

**Referee comment 3.**

The methodology section is also quite long and confusing. It will be useful to use one of the cases as an example to illustrate the methodology.

**Authors reply 3.**

To the suggestions of the referee we have revised section 2.3 and we have reorganized the results section with one example, event 14, to better illustrate the methodology. We have added **Section 3.1 "case study illustration"** which contains one event of a synergic use of the analysis method. This section includes the results from: one MODIS successful retrieval (event 14, ocean retrieval), one CALIPSO retrieval (event 14) and, one MODIS unsuccessful retrieval (event 13, land retrieval).

**Section 2.3 "Synergic approach"** has been reformulated and reduced for avoiding any confusing text. The section at lines 240 - 293 now reads:

[revised manuscript text omitted]

*have on current satellite retrieval capabilities.* 163

164

**Referee comment 4.** 165

From Sections 3.4 and 4, it is concluded that there is no agreement between the different approaches and even 166

with the literature. Even though the differences are explained, how can you validate the method you propose in 167

this paper? What is your reference? In this section, it is necessary to include the uncertainties in order to make the 168

comparison. 169

170

**Authors reply 5.** 171

There is no "**overall**" agreement between the different approaches and the cited literature. The differences are 172

explained here to better highlight the complex nature of these cases and the wide range of values that could be 173

obtained from different conditions, events and sensors. This section also supports in detailing some of the 174

limitations presented by this method. However we agree that the first draft of **section 3.4** does a poor job of 175

explaining which reference studies agree and which do not agree well with the findings in this paper. To this 176

extent we revised **section 3.4**, now **section 3.5**, and **section 4** to indicate the mentioned similarities and 177

differences. **Section 3.5** is now **Data comparison between methods and other similar studies.** This section has 178

been extensively revised to better illustrate how the methods compare to one another and to similar studies. 179

Uncertainty intervals have been added to our results and, to the extent of which they were addressed in similar 180

studies, uncertainty intervals were also added to the reference values in table 8: 181

182

183

*Table 8. Oil smoke optical properties from ground based and flight measurements along with the scientific reference.* 184

| Reference | AOD 532 nm | AOD 1064 nm | LIDAR | | | |
|---|---|---|---|---|---|---|
| | | | AE 550/1064 nm | PDR 532 nm | LR 532 nm (sr) | LR 1064 nm (sr) |
| This study CALIPSO | $0.025 \pm 0.010 - 1.526 \pm 0.804$ | $0.023 \pm 0.017 - 1.430 \pm 0.473$ | $-0.03 - 0.39$ | $0.11 \pm 0.43 - 0.32 \pm 0.48$ | $37 \pm 15 - 109 \pm 47$ | $37 \pm 15 - 86 \pm 10$ |
| (Okada et al., 1992) Ground based lidar | - | - | - | $0.14 - 0.18$ | - | - |
| (Ross et al., 1996) Airborne lidar | 0.2 - 0.6 | - | - | - | 38 | - |
| (Laursen et al., 1992) Airborne lidar | $0.05 - 1 \pm 65\%$ | $0.05 - 1.2 \pm 85\%$ | - | - | - | - |
| (Ceolato et al., 2020) Ground based lidar | - | - | - | 0.058 | - | - |
| (Ceolato et al., 2021) Ground based lidar | - | - | - | - | 125.3±5.0 sr | - |

185

| Reference | Radiometer | | | Sun photometer | | |
|---|---|---|---|---|---|---|
| | AOD 550 nm | AE 550/860 nm | $R_{eff}$ (μm) | AOD 500 nm | AE 440/870 nm | $R_{eff}$ (μm) |
| This study MODIS and AERONET | - 0.04 – 0.16 ±(0.05 + 0.20 × AOD) | - 0.18 – 1.25 | 0.29 – 1.73 μm | 0.28 – 0.68 ± 0.01 | 0.45 – 0.90 | - |
| (Pilewskie and Valero, 1992) Airborne radiometer | 0.82 – 1.92 ± 2% (500 nm) | - | - | - | - | - |
| (Nakajima et al., 1996) | - | - | - | 1.5 | 0.7 ± 2.5 % | - |
| (Mather et al., 2007) | | | | 0.3 – 1.6 (440 nm) | 0.09 – 0.42 | 0.45 – 1.40 μm |

186

Please keep in mind that not all studies in table 8 presented estimated uncertainty intervals. We directly compared our CALIPSO results to studies which utilized lidar measuring techniques, either airborne or ground based. Similarly we compared our MODIS and AERONET data to similar studies utilizing sun photometers or air borne radiometers (only one study was found). Please find the revised section 3.5 at lines 560 – 621 and section 4 at lines 632 – 661.

**Referee comment 5.**

193

Why didn't you use SSA? The analysis of the SSA (or the AAOD) will add a great value to the study since one of the interests of studying smoke lies on its absorbing capacity. Data from a different sensor, such as OMI, could be of interest.

194
195
196
197

**Authors reply 5.**

198

We agree that these properties would have been of great interest and value to our study and we would like to assure the referee that the potential of adopting SSA and AAOD was thoroughly considered and investigated. Unfortunately there were no overlapping SSA or AAOD pixels from the OMI sensor which would correspond to any of the case studies found. The OMI product would also not be of much statistical relevance as the pixel size is too large and the retrieval would effectively describe a large contribution of the local background values and not so much the smoke plumes themselves. Other similar products, such as Tropomi, may have smaller pixel size products but they lack the temporal coverage. As current satellite missions progress and advancements to sensor retrievals develop over time, we express our commitment to further analyse smoke plumes to better understand these types of aerosols.

**Specific remarks**

209

**Referee comment 1.**

210

Line 265: What is the information obtained from the analysis of the unsuccessful retrievals? It is useful for the study or even reliable?

211
212
213

**Authors reply 1.**

214

The cases that are found to have unsuccessful retrievals are significant in highlighting the limitations of MODIS sensor when judging these smoke plumes. However the unsuccessful retrievals are not used to extract plume aerosol properties, and to this extent we agree that the discussions from lines 470 – 496 can be reduced. In this

215
216
217

regard we kept one unsuccessful retrieval to discuss in detail and to better highlight the methodology. This case 218
is now part of section 3.1 at lines 336 – 349 and now reads: 219

*Figure 3 shows an example of an unsuccessful retrieval of the land algorithm for the event 13 plume on* 220
*30.12.2014. We can distinguish the plume from the RGB image over the Gulf of Sidraa while also observing AOD* 221
*values over land where the smoke plume drifted E-NE towards the island of Crete. However, there seems to be no* 222
*distinguishable AOD gradient, over land, in the plume section. A further inspection suggested that all pixels* 223
*showed values of 0.095 which suggest that the lower radiance values did not match well with pre-existing LUT* 224
*values. Consequently, the region is classified as "clean atmosphere" and thus, a unique AOD value is assigned* 225
*to all the pixels. Conversely, the ocean algorithm retrieved AOD that varied between 0.1 and 0.37. Since these* 226
*heavy smoke plumes are the result of extreme scenarios they are rarely observed and may not end up being a* 227
*subject of research. Thus, we believe there are no cases within the LUT values describing extremely low* 228
*atmospheric transmission and radiance values, highly absorbent aerosol, low SSA and low reflectance values* 229
*over a large spectral range including MODIS bands 1 through 7.* 230

231

[Figure]

232

*Figure 3. Retrieval of plume (unsuccessful) and background AOD values: event 13, 30.12.2014. The red coloured "x"* 233
*indicates the event origin.* 234

**Referee comment 2.** 235
Tables 2, 3, 4 and 5: A column with the name of the oil fires will make them easier to identify in the tables. 236

237

**Authors reply 2.** 238
All tables describing plume events have been updated with a column containing the "event ID no." as suggested 239
by referee 2, and in text we have addressed each event by this number for consistency reasons. This column is 240
added to tables 2 to 7. 241

242

**Referee comment 3.** 243
Figure 3: Use the same scale for the AOD to ease the comparison among the different figures (this comment can 244
be applied to all the figures) 245

246

247

248

**Authors reply 3**

We have modified all AOD figures to which this scale applies. Formerly know figures 3, 4, and 5 have been merged with respect to the events discussed in detail. The merger has resulted in figure 7. Please find the revised figures bellow:

[Figure]

Figure 2. Visual representation of the analysis method for MODIS data: (a) - plume captured in true color; (b) - AOD retrieval over the plume area and background (Gulf of Sidra); (c) - AOD retrieval as a result of plume minus background values; (d) – Angstrom exponent for plume and background area; (e) – Effective radius for plume and background area. The red coloured "x" indicates the event origin.

[Figure]

Figure 7. (a) Successful retrievals of aerosol properties for events 13 and 16. Plume specific AOD; (b) AE values for plume and the local background; (c) $R_{eff}$ values for plume and the local background. The red coloured "x" indicates the event origin.

**Referee comment 4.**

Line 385: You indicate that "This is evident in the plume albedo from MODIS true colour images.", but the RGB images are not included. Include them or rephrase the sentence.

**Authors reply 4.**

After reducing the overall results section, this sentence is no longer found in the revised manuscript.

**Referee comment 5.**

Line 507: In figure 7b there are no data below 3150 m, how do you identify the plume base and top?

**Authors reply 5.**

We used the Particulate backscatter coefficient (532 nm) 5 km Aerosol Profile to determine the layer top and base. The values are cut off below 3150 m, for this specific backscatter profile, due to the sensitivity of the backscattered signal being reduced or lost because of the strongly attenuated two-way transmission. This case was treated by CALIPSO as an opaque aerosol layer. In this case the same values for the plume top and base, are also found in the variables "Layer_base_altitude" and Layer_top_altitude" part of the 5 km Aerosol Layer product. Lines 353

– 355 now read: *The average plume thickness was approximately 920 m. The layer base was situated between*    280
*2600 and 3100 m above the Gulf while the top was measured between 3300 and 4200 m*.    281

    282

**Referee comment 6.**    283

Line 553: What does imply for this study that the SIBYL algorithm failed to detect the plume area and level 2    284
products averaged 20 km were used? Is the information obtained accurate for the study of the smoke plume?    285

    286

**Authors reply 6.**    287

Based on the specific conditions of the 29[th] of December 2014 at As Sidra we concluded that the low background    288
AOD values in conjunction with the narrow plume section led to a larger averaging scheme, from 5 km to 20 km.    289
There is a discernible difference between the background and plume section. In this sense the information does    290
reflect the presence of additional aerosols in the study area. However due to this larger averaging scheme we    291
suspect that these values represent a more conservative estimate as opposed to the other case studies where the    292
averaging scheme was done at 5 km. The sentence was rephrased and now reads: *The SIBYL algorithm level 2*    293
*products were averaged over a larger 20 km area, as opposed to the 5 km averaging resolution, thus plume values*    294
*are harder to distinguish from background aerosol levels.* Lines 452 – 454.    295

---

## Author Comment (AC2)

**General remarks** 1

**Referee comment 1.** 2

In my opinion the authors should improve the organization of the paper and help the reader in 3
understanding the rationale of some choices made. For example, what's the reason for the order in 4
which the events are listed in table 1 (certainly not chronological)? I would suggest that the events 5
discussed in detail in the text are referred to by their numbering in the table; otherwise it is sometimes 6
difficult to understand which event the authors are discussing, especially because sometimes they allude 7
to them with a name different from the location given in table 1. For example, in line 133 it is said that 8
"only the event in Kiev was analysed by different techniques". One has to guess that the referred event 9
is number 14 in the actual numbering of table 1. The Caspian sea event (No. 8 in table 1) is also referred 10
to as the Gunashli event (line 435); the reader needs to be a geography expert to realize that the authors 11
are talking about the same event. 12

**Author reply 1** 14

To address the first question regarding the apparent order of the events listed in table 1: originally there 15
was no obvious reason for the order of the events; the event order in table 1 merely reflects the order in 16
which the authors identified these events in literature. We kept this order out of consistency throughout 17
the study as the analysis was carried out. We agree that the preprint version of table 1 was confusing 18
and as such, we made the following adjustments: 19

- We added the column "*event ID no.* ", as suggested by the referee, and in text we addressed 20
each event by this number for consistency reasons; 21

- The former columns entitled "Date" and "Number of observations (days)" have been merged 22
into a single column "*MODIS observation interval*". This new column contains the dates for the "*first* 23
*day*" and the "*last day*" in which MODIS RGB images showed oil smoke plumes. Some events, i.e. 24
event 3 and 14, are listed as having more than one location. We have grouped all locations within a 25
single event because they share the same cause (same armed conflict) which generated the events on 26
the same "*first day*". For these events, the coordinates for each location (oil installation involved in a 27
fire) in particular are given. 28

Please find the revised version of table 1 below: 29

30

*Tab. 1 Major industrial events leading to observable smoke plumes seen in MODIS RGB images* 31

| Event ID No. | Location | MODIS observation interval | | Coordinates | Cause of event | Type of installation | References |
|---|---|---|---|---|---|---|---|
| | | Start | End | | | | |
| 1 | Qayyara, Iraq | 13.06.2016 | 27.03.2017 | 35.83 N ; 43.21 E | armed conflict | oil wells | (Tichý and Eichler, 2018) |
| 2 | Omidieh, Iran | 06.05.2019 | 06.05.2019 | 30.84 N ; 49.65 E | human error | oil pipeline | (Financial Tribune, 2019) |
| 3 | Haradh, Hawiyah, Uthmaniyah, Shedgum, Buqayq; Saudi Arabia | 14.09.2019 | 26.09.2019 | 24.05 N ; 49.20 E 24.80 N ; 49.35 E 25.18 N ; 49.31 E 25.64 N ; 49.39 E 25.92 N ; 49.68 E | armed conflict | oil processing | (Khan and Zhaoying, 2020) (Reuters, 2019) (New York Times, 2019) |
| 4 | Caspian Sea, Azerbaijan | 06.12.2015 | 18.12.2015 | 40.20 N ; 51.06 E | extreme weather | oil and gas platform | (Necci et al., 2019) |
| 5 | Gulf of Mexico, USA | 21.04.2010 | 21.04.2010 | 28.44 N ; 88.21 W | equipment failure | drilling rig | (Gullett et al., 2016) |
| 6 | East China Sea, China | 14.01.2018 | 14.01.2018 | 28.37 N ; 126.08 E | human error | oil tanker | (Li et al., 2019) (Qiao et al., 2019) |
| 7 | Houston Texas, USA | 18.03.2019 | 19.03.2019 | 29.43 N ; 95.05 E | equipment failure | storage tanks | (An Han et al., 2020) |

| | | | | | | | |
|---|---|---|---|---|---|---|---|
| 8 | Jaipur, India | 30.10.2009 | 08.11.2009 | 26.77 N ; 75.83 E | Human error | storage tanks | (Vasanth et al., 2014) |
| 9 | Sendai, Japan | 12.03.2011 | 13.03.2011 | 38.27 N ; 141.03 E | earthquake, tsunami | storage tanks | (Krausmann and Cruz, 2013) |
| 10 | Vasylkiv, Ukraine | 09.06.2015 | 10.06.2015 | 50.16 N ; 30.32 E | sabotage | storage tanks | (Kovalets et al., 2017) (Reuters, 2015) |
| 11 | Ra's Lanuf, Libya | 19.08.2008 | 25.08.2008 | 30.45 N ; 18.49 E | human error | storage tanks | (The Telegraph, 2011) |
| 12 | Ra's Lanuf, Libya | 12.03.2011 | 14.03.2011 | 30.45 N ; 18.49 E | armed conflict | storage tanks | (BBC, 2011) |
| 13 | As Sidr, Libya | 26.12.2014 | 31.12.2014 | 30.60 N ; 18.28 E | armed conflict | storage tanks | (BBC, 2014) |
| 14 | Ra's Lanuf, As Sidr; Libya | 05.01.2016 | 07.01.2016 | 30.45 N ; 18.49 E 30.60 N ; 18.28 E | armed conflict | storage tanks | |
| 15 | Surt disrtric, Libya | 14.01.2016 | 14.01.2016 | 30.02 N ; 18.50 E | armed conflict | oil pipeline | (Tichý and Eichler, 2018) (Tichý, 2019) |
| 16 | Ra's Lanuf, Libya | 21.01.2016 | 23.01.2016 | 30.45 N ; 18.49 E | armed conflict | storage tanks | |
| 17 | Ajdaviya district, Libya | 01.02.2016 | 01.02.2016 | 29.68 N ; 20.54 E | armed conflict | oil pipeline | |
| 18 | Ra's Lanuf, Libya | 17.06.2018 | 21.06.2018 | 30.45 N ; 18.49 E | armed conflict | storage tanks | (Reuters, 2018) |
| 19 | Puebla, Mexico | 19.12.2010 | 19.12.2010 | 18.96 N ; 98.45 W | illegal tapings | oil pipeline | (Biezma et al., 2020) |
| 20 | Escravos, Nigeria | 04.01.2018 | 05.01.2018 | 5.45 N ; 5.35 E | bush fire | oil pipeline | (Bloomberg, 2018) |
| 21 | Puerto Sandino, Nicaragua | 18.08.2016 | 19.08.2016 | 12.18 N ; 86.75 W | unknown | storage tanks | (Ahmadi et al., 2020) |
| 22 | Gulf of Oman | 13.06.2019 | 13.06.2019 | 25.39 N ; 57.38 E | armed conflict | oil tanker | (BBC, 2019) |
| 23 | Catano, Puerto Rico | 23.10.2009 | 24.10.2009 | 18.41 N ; 66.13 W | human error | storage tanks | (Vasanth et al., 2014) |
| 24 | Punto Fijo,Venezuela | 27.08.2012 | 27.08.2012 | 11.74 N ; 70.18 W | equipment failure | storage tanks | (Schmidt et al., 2016) |
| 25 | Butcher Island, India | 07.10.2017 | 08.10.2017 | 18.95 N ; 72.90 E | lightning strike | storage tank | (The Indian Express, 2017) |

As a result, the former statement in line 133 "only the event in Kiev was analysed by different techniques" now reads as "*only event 10 was analysed by different techniques*", now at line 125.

The event formally referred to as "The Caspian Sea event" or "Gunashli event" is now only referred to as "*event 4*".

To further address the improvements in the organization of the paper we rearranged the results section, as suggested by referee 2, to better illustrate the analysis method and the overall results. The current format of the results section includes:

Section 3.1 **Case study illustration** – we examined one event to illustrate the analysis method. This section includes the results from: one MODIS successful retrieval (event 14, ocean retrieval), one CALIPSO retrieval (event 14) and, one MODIS unsuccessful retrieval (event 13, land retrieval).

Section 3.2 **MODIS successful retrievals**

In lines 298 – 303 we address how many successful retrievals were analysed:

"*Based on the information given in table 1 we filtered a total of 375 days in which oil smoke plumes were observed by the MODIS sensors. After applying the selection criteria for the MODIS sensor we obtained a total of 10 days with successful retrievals. The majority of oil plumes resulted in unsuccessful retrievals, 70.7%, while 26.7% of plumes were screened out due to high percentage of cloud coverage. When applying the selection criteria for CALIPSO we obtained a number of 6 plume sections suitable for analysis. Table 2 shows the dates for both MODIS and CALIPSO retrievals suitable for analysis.*"

A list of MODIS successful retrievals is given in table 2 as follows:

*Table 2. List of successful MODIS retrievals and CALIPSO overpass dates*

| Event Id. Nr. | MODIS (Terra and Aqua) Successful retrieval date | CALIPSO retrieval date |
|---|---|---|
| 1 | - | 01.07.2016 17.07.2016 21.10.2016 |
| 4 | 08.12.2015 | - |
| 5 | 21.04.2010 | - |
| 9 | 11.03.2011 | - |
| 11 | - | 22.08.2008 |
| 13 | 28.12.2014 29.12.2014 30.12.2014 | - 29.12.2014 - |
| 14 | 06.01.2016 | 06.01.2016 |
| 16 | 21.01.2016 | - |
| 20 | 19.08.2016 | - |
| 21 | 04.01.2018 (only Aqua) | - |

Section 3.2 shows general MODIS results together with a more detailed discussion of the events in Lybia, event 13 and 16 (event 14 was previously discussed in section 3.1). The reasoning behind the more detailed discussion of these events is given in line 423: "*We choose to describe in detail the events from Libya as they are also analysed based on CALIPSO retrievals*"

Section 3.3 shows general results from **CALIPSO retrievals** together with detailed discussion of the events in Lybia.

Section 3.4 is now the **AERONET case study** and,

Section 3.5 is now **Data comparison between methods and other similar studies.** This section has been extensively revised to better illustrate how the methods compare to one another and to similar studies. Uncertainty intervals have been added to our results and, to the extent of which they were addressed in similar studies, uncertainty intervals were also added to the reference values in table 8:

*Table 8. Oil smoke optical properties from ground based and flight measurements along with the scientific reference.*

| Reference | AOD 532 nm | AOD 1064 nm | LIDAR | | | |
| | | | AE 550/1064 nm | PDR 532 nm | LR 532 nm (sr) | LR 1064 nm (sr) |
|---|---|---|---|---|---|---|
| This study CALIPSO | 0.025 ± 0.010 – 1.526 ± 0.804 | 0.023 ± 0.017 - 1.430 ± 0.473 | - 0.03 – 0.39 | 0.11 ± 0.43 - 0.32 ± 0.48 | 37 ± 15 - 109 ± 47 | 37 ± 15 - 86 ± 10 |
| (Okada et al., 1992) Ground based lidar | - | - | - | 0.14 – 0.18 | - | - |
| (Ross et al., 1996) Airborne lidar | 0.2 - 0.6 | - | - | - | 38 | - |
| (Laursen et al., 1992) Airborne lidar | 0.05 – 1 ± 65% | 0.05 – 1.2 ± 85% | - | - | - | - |
| (Ceolato et al., 2020) Ground based lidar | - | - | - | 0.058 | - | - |
| (Ceolato et al., 2021) Ground based lidar | - | - | - | - | 125.3±5.0 sr | - |

| Reference | Radiometer | | | Sun photometer | | |
|---|---|---|---|---|---|---|
| | AOD 550 nm | AE 550/860 nm | $R_{eff}$ (μm) | AOD 500 nm | AE 440/870 nm | $R_{eff}$ (μm) |
| This study MODIS and AERONET | - 0.04 – 0.16 ±(0.05 + 0.20 × AOD) | - 0.18 – 1.25 | 0.29 – 1.73 μm | 0.28 – 0.68 ± 0.01 | 0.45 – 0.90 | - |
| (Pilewskie and Valero, 1992) Airborne radiometer | 0.82 – 1.92 ± 2% (500 nm) | - | - | - | - | - |
| (Nakajima et al., 1996) | - | - | - | 1.5 | 0.7 ± 2.5 % | - |
| (Mather et al., 2007) | | | | 0.3 – 1.6 (440 nm) | 0.09 – 0.42 | 0.45 – 1.40 μm |

82
83
84
85

**Referee comment 2.** 86

Which events have been studied in a really synergic (MODIS + CALIPSO) way? Only one? ("The only 87 event which was captured by both MODIS and CALIPSO retrievals showed a large level of 88 discrepancy", lines 698-699). If this is so, I would strongly suggest that the title is modified. Both the 89 MODIS-based and the CALIPSO-based analyses are worth being presented, but in that case the 90 "synergic" term in the title would be too strong in my view. 91

92

**Authors reply 2.** 93

The statement at lines 698 – 699: "The only event which was captured by both MODIS and CALIPSO 94 retrievals showed a large level of discrepancy" needs further clarification. This event, event 14, is the 95 only event which was captured by both sensors (MODIS and CALIPSO) at approx. 2 minutes apart. 96 Event 13 was retrieved by CALIPSO at nighttime within a 12 hour interval from the closest MODIS 97 retrievals. In this specific case the CALIPSO retrieval 29.12.2014 at approx.00:30 UTC sits between 98 the MODIS successful retrievals on 28.12.2014 at 09:30 UTC (Terra) and 12:40 (Aqua), and on 99 29.12.14 at 10:10 (Terra) and 11:45 (Aqua). To some extent this event is also seen by both sensors 100 (within approx. 12 hour time discrepancy) although we agree that this is may not be enough for full 101 synergy. The remaining events are either successful MODIS retrievals with no overlapping CALIPSO 102 retrievals or CALIPSO retrievals with unsuccessful (over land) MODIS retrievals. The narrow swath 103 width and rare revisiting interval of CALIPSO are the main issues for the low number of overlapping 104 acquisitions. In the rare cases when retrievals overlap, the smoke plumes need to satisfy the selection 105 criteria and to be the result of successful retrievals. We agree that to some extent the title may be 106 misleading as most of these events are not overlapped by both sensors. As such, we agree that "synergic" 107 in the title may not be appropriate. To this extent we revised the title which now reads: *A novel method* 108 *of identifying and analysing oil smoke plumes based on MODIS and CALIPSO satellite data.* The 109 statement at lines 698 – 699 now reads: "*The only event which was captured by both MODIS and* 110 *CALIPSO retrievals, within 2 minutes apart, showed a large level of discrepancy."* 111

112
113

**Referee comment 3.** 114

The rationale after which some events are discussed in particular should be given to help the reader understand the reason why these events, and not others, are discussed. For example, the case of Ra's Lanuf and As Sidr is properly introduced ("We selected a successful retrieval to better describe the method used for our analysis", line 342), as it is the SOCAR Platform in the Caspian Sea ("Next, we examined the smoke plume from SOCAR's Platform No.10 fire in the Caspian Sea as an "atypical" event based on the fuel type and plume albedo", lines 374-375), but the reason why the next events described in section 3.1 (Deepwater Horizon, JX Nippon, Ra's Lanuf again, Puerto Sandino, Escravos…) are chosen is unclear.

**Authors reply 3.**

In table 1 we describe all the events that were visible in MODIS RGB images. What we failed to mention is which of these events remained as successful or unsuccessful retrievals and which did not qualify as neither, nor getting passed the selection criteria. To this extent we added the following paragraph and list of successful retrievals.

In lines 298 – 303 we address how many successful retrievals were analysed:
"*Based on the information given in table 1 we filtered a total of 375 days in which oil smoke plumes were observed by the MODIS sensors. After applying the selection criteria for the MODIS sensor we obtained a total of 10 days with successful retrievals. The majority of oil plumes resulted in unsuccessful retrievals, 70.7%, while 26.7% of plumes were screened out due to high percentage of cloud coverage. When applying the selection criteria for CALIPSO we obtained a number of 6 plume sections suitable for analysis. Table 2 shows the dates for both MODIS and CALIPSO retrievals suitable for analysis*.*"

A list of MODIS successful retrievals is given in table 2 as follows:

*Table 2. List of successful MODIS retrievals and CALIPSO overpass dates*

| Event Id. Nr. | MODIS (Terra and Aqua) Successful retrieval date | CALIPSO retrieval date |
|---|---|---|
| 1 | - | 01.07.2016 17.07.2016 21.10.2016 |
| 4 | 08.12.2015 | - |
| 5 | 21.04.2010 | - |
| 9 | 11.03.2011 | - |
| 11 | - | 22.08.2008 |
| 13 | 28.12.2014 29.12.2014 30.12.2014 | - 29.12.2014 - |
| 14 | 06.01.2016 | 06.01.2016 |
| 16 | 21.01.2016 | - |
| 20 | 19.08.2016 | - |
| 21 | 04.01.2018 (only Aqua) | - |

In the initial draft of the paper we addressed in detail all the MODIS successful retrieval events and a selection of unsuccessful retrievals. At the suggestion of referee nr.2 we reduced the overall length of the results section, discussing in detail only the events in Lybia, events 11, 13, 14, 16. In the revised

manuscript we added Lines 422 – 423 which reflect this reasoning: *We choose to describe in detail the* 146
*events from Libya as they are also analysed based on CALIPSO retrievals. Moreover the plumes* 147
*resulting from these events share the same locations (As Sidr and Ra's Lanuf).* 148

149

**Main specific remarks** 150
**Referee comment 1.** 151

152

I think that the abstract is too specific in giving values of AOD and AE obtained from MODIS and 153
CALIPSO and discussing the discrepancies between MODIS and CALIPSO retrievals for the (yet 154
unidentified in the abstract) event of Ra's Lanuf on 6th of January 2016. Although this is debatable, in 155
my opinion the abstract should highlight the contributions of the method, rather than the results 156
obtained. 157

158

**Author reply 1** 159
Based on this suggestion we have revised the abstract to better reflect the contributions of the method 160
and the overall findings of the study. Lines 23 – 37 reflect these changes: 161

162

*The analysis method in this study was developed to better differentiate between oil smoke aerosols and the local* 163
*atmospheric scene. We present several aerosol properties in the form of plume specific averaged values. We* 164
*believe that MODIS values are a conservative estimation of plume AOD since MODIS algorithms rely on general* 165
*aerosol models and various atmospheric conditions within the look-up tables which do not reflect the highly* 166
*absorbing nature of these smoke plumes. Based on this study we conclude that the MODIS land algorithms are* 167
*not yet suited for retrieving aerosol properties for these types of smoke plumes due to the strong absorbing* 168
*properties of these aerosols. CALIPSO retrievals rely heavily on the type of lidar solutions showing discrepancy* 169
*between constrained and unconstrained retrievals. Smoke plumes identified within a larger aerosol layer were* 170
*treated as unconstrained retrievals and resulted in conservative AOD estimates. Conversely, smoke plumes* 171
*surrounded by clear air were identified as opaque aerosol layers and resulted in higher lidar ratios and AOD* 172
*values. Measured lidar ratios and particulate depolarization ratios showed values similar to the upper ranges of* 173
*biomass burning smoke. Results compare well with studies that utilized ground-based retrievals, in particular for* 174
*Ångström exponent (AE) and effective radius ($R_{eff}$) values. MODIS and CALIPSO retrieval algorithms disagree* 175
*on AOD ranges, for the most part, due to the extreme light absorbing nature of these types of aerosols. We believe* 176
*that these types of studies are a strong indicator for the need of improved aerosol models and retrieval algorithms.* 177

178

**Referee comment 2.** 179
In section 1.2 (Event synopsis), why only a few events are discussed? What's the rationale for their 180
selection? 181

182

183

**Authors reply 2.** 184
Section 1.2 is reserved only for the events in Libya and the events in Iraq. These include the events 185
discussed in detail in sections 3.1, 3.2, 3.3 and 3.5. The authors felt that extending the events synopsis 186

to include all the events would make a long paper containing not so relevant details to the overall analysis.

**Referee comment 3.**

The sentence in lines 302-304 sounds confusing: "Focusing on the plume bins we used the extinction coefficient to backscatter ratio to determine the plume averaged lidar ratios if the plume was not identified as a distinct aerosol feature." But the extinction coefficient to backscatter [coefficient] ratio is the lidar ratio. The explanation looks circular.

**Authors reply 3.**

We agree that the sentence sounds confusing and repetitive. Section 2.3 has suffered extensive revision and the line referring to the lidar ratios now reads: *Additionally, the plume extinction-to-backscatter (i.e., lidar ratio), Ångström (532/1064 nm) exponent, and particle depolarization ratio are assessed to investigate the type-dependent characteristics of the plume and whether oil smoke presents distinctive intensive properties",* now at lines 287 – 289.

**Referee comment 4.**

The events in table 2 and 3 are difficult to identify based only on the date. Note in addition that in table 3 there is a row more than in table 2. Didn't MODIS in Terra retrieve values for the event of 04.01.2018? The same remark on the inconvenience to identify events based on the date apply to tables 4 and 5. Moreover, in these tables the events coincident with events in table 2 and 3 should be pointed out. The event on 21.10.16 doesn't seem to be listed in table 1. Is the date correct? Should it be 21.01.16?

**Authors reply 4.**

As discussed in the **general remark nr. 1**, we have added a "*MODIS observation interval"* which contains the dates for the "*first day"* and the "*last day"* in which MODIS RGB images showed oil smoke plumes. This effectively means that MODIS has "seen" oil smoke plumes in that specific time interval, however not all of the dates in the said interval may result in successful retrievals. As such we added table 2 – "*List of successful MODIS retrievals and CALIPSO overpass dates"* to show the exact dates for successful MODIS retrievals and CALIPSO retrievals. Going forward, these dates are found in the revised manuscript in tables 3 to 7. Please keep in mind that we also added the event ID number to above mentioned tables in order to better identify the plumes in question. Event 21, on 04.01.2018 was only retrieved by Aqua due to sun glint. The date 21.10.16 refers to a CALIPSO retrieval from event 1 while the date 21.01.16 refers to a MODIS retrieval from event 16. This is made clear in the revised manuscript by adding the information in table 2.

**Referee comment 5.**

Apart from the issues mentioned in point 3, I had a lot of trouble following section 3.1. The initial discussion (lines 342 – 362) refers to figure 2. However, in lines 362 – 363, tables 2 and 3 are referred without saying that they include data from other locations, described in figure 3 and in the subsequent text. In addition, figure 3c is quoted twice ("Figure 3c shows a plume specific AOD values 390 ranging

from 0.06to 0.23", lines 390-391; "Figure 3d shows AOD values as high as 0.24 over the average AOD    229
background level for the plume originating at Ra's Lanuf.", lines 391-392), but the second time should    230
be probably figure 3d. Then figure 3d is cited again in line 394. It would be better to group together all    231
the discussion referred to given event. One has to guess, moreover, that the Ra's Lanuf oil field pertains    232
(probably, not sure) to the Surt district of Lybia (line 394).    233
     234

**Authors reply 5.**    235
**Section 3.1** has also suffered extensive revision and is now **section 3.2**. Lines 395 – 415 now refer to    236
the overall results of the MODIS successful retrievals. As such, the results from tables 3 and 4 are    237
discussed on a more general term. A more detailed discussion portraying the events in Libya (event 13    238
and 16 since event 14 was previously discussed in section 3.1) is found in lines 421 – 439. Figure 7 and    239
the discussion that stem from it are presented below:    240
     241

*Following event 14 in figure 2, figure 7 shows a visual representation of MODIS successful retrievals*    242
*from events 13 and 16. We choose to describe in detail the events from Libya as they are also analysed*    243
*based on CALIPSO retrievals. Moreover the plumes resulting from these events share the same*    244
*locations (As Sidr and Ra's Lanuf). Figure 7a shows plume specific AOD values ranging from 0 to 0.28.*    245
*Plumes from As Sidr, event 13, are visible in the first three rows of figure 7. This event was captured in*    246
*multiple days while the fire engulfed several oil tanks and subsequently injected higher amounts of*    247
*aerosols in the region. Depending on the local background levels, average plume specific AOD ranged*    248
*from -0.03 to 0.15. Negative values can be explained by the presence of dust and marine aerosols in the*    249
*atmospheric background. This is especially evident for event 13 on 30.12.2014 when high background*    250
*levels were registered in the Gulf of Sidra while lower levels were seen off the shores of At Tamimi, 600*    251
*km NE of As Sidr. The fourth row in figure 7 shows the plume from event 16, marking the second attack*    252
*on the Ra's Lanuf tank farm in 2016. The plume section over the Gulf recorded AOD values twice as*    253
*high as the background level however the net contribution amounted, on average, to a value of 0.10.*    254
*The AE values below 0 seen in 7b suggest a coarse dominant scene. Figure 7b also shows low AE values*    255
*identified further from the plumes edge showing the spatial extent of these types of aerosols. The Gulf*    256
*of Sidra is situated in one of the main pathways of long range transported dust (Kallos et al., 2007)*    257
*thus affecting AE local background values, as seen in table 3 and table 4. In figure 7c we identify high*    258
*$R_{eff}$ values consistently over 1 μm while, in some cases, values close to the fire and within the center of*    259
*the plume area reached the maximum 2.50 μm. These large values are consistent with the observed AE*    260
*trend observed indicating larger particles and coarse mode dominant aerosol type. Background values*    261
*for these events fluctuated between 0.32 and 1.04 μm due to regional dust-like aerosols.*    262

[Figure]

Figure 7. (a) Successful retrievals of aerosol properties for events 13 and 16. Plume specific AOD; (b) AE values 264
for plume and the local background; (c) $R_{eff}$ values for plume and the local background. The red coloured "x" 265
indicates the event origin. 266

267

268

**Referee comment 6.** 269

In figure 3, the color scale seems to correspond to the plume specific AOD. This should be indicated in 270
the legend of the color scale to avoid confusion with the total AOD. Please check also what is 271
represented in the color scales of figure 6. 272

273

**Authors reply 6.** 274

The former figures 3, 4 and 5 were merged into one figure (figure 7 presented above) after the 275
suggestion form referee 2 to reduce the overall length of the results section and the overall number of 276
figures. Please keep in mind that figure 7 now only refers to events 13 and 16 as a result. Figure 7.a 277
color scale now indicates the plume specific AOD. 278

279

**Referee comment 7.** 280

Perhaps related to the last part of point 5 above, an event in As Sidr is mentioned in table 1 (No. 19), 281
but with no date. Is it the same event as No. 20? 282

**Authors reply 7.**

This has been clarified in the revised version of table 1. Formerly the numbering was done for the locations and not for each individual event. By adding the event ID no. the events and their locations are much clearer. The former No.19 is part of event 14 while the former No. 20 is part of event 15 as seen in the revised version of table 1.

**Referee comment 8.**

Referring to figure 6, it is said (line 476) that "The cases form Saudi Arabia, Iran and Iraq show no values retrieved over the plume areas". What's the reason to say that in the case of panels a and b? Where should the reader expect the plume? By the way it would be helpful to say that these cases correspond to figures 6a, 6b and 6c.

**Authors reply 8.**

Referring to the former figure 6, the cases form Saudi Arabia, Iran and Iraq were indeed shown in figures 6a, 6b and 6c. In the center of each image one could observe the black smoke plumes in question. This would also correspond to the section of the image where AOD values were not available. The section concerning the unsuccessful retrievals, formerly 469 – 495, has been overall reduced, as suggested by referee 2 since it was not directly relevant to the analysis. In the revised manuscript we discuss one unsuccessful retrieval, lines 336 – 345 and shown in figure 3, to better describe the limitation of the MODIS land algorithm.

*Figure 3 shows an example of an unsuccessful retrieval of the land algorithm for the event 13 plume on 30.12.2014. We can distinguish the plume from the RGB image over the Gulf of Sidraa while also observing AOD values over land where the smoke plume drifted E-NE towards the island of Crete. However, there seems to be no distinguishable AOD gradient, over land, in the plume section. A further inspection suggested that all pixels showed values of 0.095 which suggest that the lower radiance values did not match well with pre-existing LUT values. Consequently, the region is classified as "clean atmosphere" and thus, a unique AOD value is assigned to all the pixels. Conversely, the ocean algorithm retrieved AOD that varied between 0.1 and 0.37. Since these heavy smoke plumes are the result of extreme scenarios they are rarely observed and may not end up being a subject of research. Thus, we believe there are no cases within the LUT values describing extremely low atmospheric transmission and radiance values, highly absorbent aerosol, low SSA and low reflectance values over a large spectral range including MODIS bands 1 through 7.*

[Figure]

319

Figure 3. Retrieval of plume (unsuccessful) and background AOD values: event 13, 30.12.2014. The red coloured 320
"x" indicates the event origin. 321

322

**Referee comment 9.** 323

Relating to the Ra's Lanuf plume on 6th of January, there are statements on the retrieved lidar ratio 324
values that should be clarified: it is said on the one hand that the lidar ratios of 109 sr at 532 nm and 86 325
sr at 1064 nm (lines 513-514 and table 6) are obtained. Then, in lines 541-543 it is stated that: "It should 326
be mentioned that this event was an optimal case for constrained lidar ratio retrieval since the feature 327
was surrounded by clear air. However, due to the fact that the plume feature was completely opaque, 328
the lidar ratio could not be obtained from a constrained solution via the two-way layer transmittance". 329
This should be clarified as, at first sight, it looks contradictory. I suggest that the CALIPSO retrievals 330
of lidar ratio on opaque layers is briefly discussed or, at least, mentioned and referenced. 331

332

**Authors reply 9.** 333

334

The statement regarding the constrained lidar ratio and the opaque layers has been clarified in lines 360 335
– 369 of the revised manuscript. We also briefly discussed how CALIPSO assessed opaque aerosol 336
layers and how this is related to the smoke plume from Ra's Lanuf 6th of January. To further clarify, 337
this event was resolved using an opaque aerosol layer scheme which is a "constrained" solution. 338
However this scheme is not the "traditional" "constrained" solution which relies on the two-way layer 339
transmittance. For opaque layers the two-way layer transmittance is considered to be zero, hence the 340
feature could not be resolved as such. Please keep in mind that this is a very rare case in which an 341
aerosol layer is treated as an opaque layer and thus may be subjected to higher levels of uncertainty. 342
We also addressed this issue in section 4, lines 647 – 653. 343

344

Lines 360 – 369: *This event is an example of an opaque aerosol layer, were the lidar did not penetrate* 345
*the plume up to the sea surface over the Gulf of Sidra. This event recorded a lidar ratio of 109 ± 47 sr* 346
*at 532 nm and 86 ± sr at 1064 nm. These values are larger than the CALIPSO V4 aerosol subtype* 347
*values for: elevated smoke 70 ± 16 (532 nm) and 30 ± 18 (1064 nm); Polluted continental/smoke 70 ±* 348
*25 (532 nm) and 30 ± 18 (1064 nm) (Kim et al., 2018). The initial lidar ratios were reduced by 5%* 349

*based on the scheme described by Young et al., 2018 for opaque aerosol layers. These events are* 350
*described as occurring infrequently (1% of all unique aerosol layers, detected in 2012; Young et al.,* 351
*2018) and may be subjected to further uncertainties. The initial value of the lidar ratio ($S_P$) is described* 352
*by Young et al., 2018 in Eq. (1). This assumes a zero value of the two-way transmittance ($T_P^2 = 0$) and* 353
*a multiple scattering factor value of 1 ($\eta = 1$). Young et al., 2018 also suggest that $\eta=1$ assumption may* 354
*not be valid for opaque aerosol layers and may introduce bias errors. These errors can be propagated* 355
*through the extinction and AOD retrievals and result in more conservative estimates.* 356

Lines 647 – 653: *In general constrained retrievals would better reflect the actual smoke properties* 357
*because they do not rely on an ad-hoc assignment of lidar ratio. However assigning a constrained* 358
*retrieval to oil smoke plumes requires 1: for the plume to be surrounded by clear air; and 2: smoke* 359
*concentrations should not exceed a threshold were total attenuation is achieved. The lidar ratios* 360
*generated from event 14 represent an extremely rare occasion where the smoke plume was treated as* 361
*an opaque aerosol layer. As such it was difficult to assess whether the lidar ratios where over or* 362
*underestimated although we believe that this current solution is still preferable to unconstrained* 363
*solutions.* 364

365

**Minor specific remarks** 366
**Referee comment 1.** 367
Line 30: "agreeance". I couldn't find that word in the dictionary. Probably the authors mean 368
"agreement". 369

370

**Authors reply 1.** 371
The statement was change with "agree" and now reads: *Results agree with studies that utilized ground-* 372
*based retrievals, in particular for Ångström exponent (AE) and effective radius ($R_{eff}$) values.* Lines 34 373
– 35. 374

375

**Referee comment 2.** 376
Lines 41-42: "CALIPSO measurements are heavily dependent on lidar ratios which are not directly 377
measured if plumes within the planetary boundary layer". The sentence sounds strange. Perhaps "are" 378
is missing after "if"? Please check. 379

380

**Authors reply 2.** 381
The overall sentence was change and it now reads: *CALIPSO retrievals rely heavily on the type of lidar* 382
*solutions showing discrepancy between constrained and unconstrained retrievals. Smoke plumes* 383
*identified within a larger aerosol layer were treated as unconstrained retrievals and resulted in* 384
*conservative AOD estimates.* Lines 29 – 31. 385

386

387

388

389

**Referee comment 3.** 390

Lines 81-82: "Most sensors cannot retrieve a wide variety of aerosol properties thus relying on inversions techniques and complex radiative transfer computations". The meaning of this sentence is unclear. Can or cannot the sensors retrieve aerosol properties? Please check.

**Authors reply 3.**

The revised sentence now reads: *Most sensors can retrieve a wide variety of aerosol properties however they relying on inversions techniques and complex radiative transfer computations.* Lines 73 - 74.

**Referee comment 4.**

In table 1, the cause of event id left blank for some events. It would be better to state "unknown" (if this is the reason for the blank space).

**Authors reply 4.**

There are no "cause" left blank in table 1 however we understand that the format of the table was not easy to follow. This is much clearer after adding the event ID number.

**Referee comment 5.**

Line 188: "construct" --- > "construction".

**Authors reply 5.**

Modified to "construction". Now at line 177.

**Referee comment 6.**

"Based on the particulate total backscatter coefficient (532 nm) we defined the plume cross section as in each range bin, the plume values were at least 2 times higher than background values". This sentence sounds strange. Please check the wording.

**Authors reply 6.**

The sentence was changed and now reads: *In this analysis, the particle backscatter coefficient is used to identify the geometrical properties of the smoke plume. The plume is defined as the area where the values are at least 2 times higher than the background, which is considered as an area of identical thickness located either above or below the plume.* Lines 283 – 285.

**Referee comment 7.**

Line 375: what does "n.d" mean in the reference "(Business-humanrights, n.d.)"?

**Authors reply 7.**

"n.d" stands for "no date".

**Referee comment 8.**

Line 385: "This is evident in the plume albedo from MODIS true colour images". A figure should either be provided or said that it is not shown.

**Authors reply 8.**

After reducing the overall results section, this sentence is no longer found in the revised manuscript.

**Referee comment 9.**

Lines 421-422: "Plume values close to 0 were retrieved near the event while average values registered two to three times lower than the local background". The sentence is not clear. Where are those values reported in the paper? Does this refer to the 0.34 and 0.74 values cited in line 419?

**Authors reply 9.**

The sentence did in fact refer to the 0.34 and 0.74 values cited in line 419. However after revising the results section, this section is no longer found.

**Referee comment 10.**

Lines 506-507: apparently referring to figure 7b it is said: "The average plume thickness was approximately 920 m ranging from 2700 m to 3300 m." But 3300 – 2700 = 600. In addition, the backscatter coefficient profile shown in fig. 7b does not go below 3100 m or so.

**Authors reply 10.**

The revised version of the manuscript reads: *Within the 15 km plume cross section we selected a particulate backscatter coefficient profile for reference, figure 4b, and based on this parameter we determine plume elevation and thickness. The average plume thickness was approximately 920 m. The layer base was situated between 2600 and 3100 m above the Gulf while the top was measured between 3300 and 4200 m.* Lines 352 – 355.

[Figure]

Figure 4 (a): CALIPSO overpass and MODIS plume contour; 7 (b): Particulate backscatter coefficient profile CALIPSO level 2 (532 nm)

**Referee comment 11.**

Lines 510-511: "average particulate backscatter (532 nm) values measured 510 0.015 km-1sr-1 while values at 1064 nm measured 0.17 km-1sr-1." The value for 1064 nm should be 0.017 km-1sr-1. Moreover, I think that the values measured for 532 nm and for 1064 nm are essentially indistinguishable, as they fall within the uncertainty interval of each other.

**Authors reply 11**.
The value was corrected from $0.17$ km$^{-1}$sr$^{-1}$ to $0.017$ km$^{-1}$sr$^{-1}$. Indeed the values fall within the uncertainty interval.

**Referee comment 12.**
Table 6: are there not uncertainty intervals for the PDRs and the lidar ratios? Note in addition that the date for the Qayyara, Iraq event (10.21.2016) is certainly wrong. Why these Qayyara events not listed in table 1? Were they not observed my MODIS?

**Authors reply 12.**
Uncertainty intervals have been added to PDRs LR and AOD in tables 6 and 7. The date was modified as follows: 21.10.2016. The Qayyara event is now listed in table 1 with the event ID number of 1. Unfortunately since this event was retrieved "over land" it did not result in any MODIS successful retrievals.

**Referee comment 13.**
Line 666: "AOD values ranged significantly…" What does this mean?

**Authors reply 13.**
The sentence has been revised and now reads: *AOD values measured between 0.02 and 1.52 and were directly influenced by fuel burning rates, local background aerosol loading and especially lidar ratio solutions.* Lines 533 – 535.

**Referee comment 14.**
Lines 680-681: "The smoke plume was also captured in RGB images as seen in figure 6g". But figure 6g does not seem to correspond to an RGB image.

**Authors reply 14.**
The sentence has been revised and now reads: *The smoke plume was also captured in RGB images as seen in figure 6, lower left image.* Lines 546 – 547.

**Referee comment 15.**
Line 683: "Figure 12d shows the daily evolution of AE with values between 0.45 and 0.9". At which wavelength?

**Authors reply 15.**

440/870 nm. The value is indicated in figure 9c and 9d.

**Referee comment 16.**

Line 709: "did not reached Kiev" → "did not reach Kiev".

**Authors reply 16.**

Revised to "reach" at line 576.